Postcranial anatomy of Pissarrachampsa sera (Crocodyliformes, Baurusuchidae) from the Late Cretaceous of Brazil: insights on lifestyle and phylogenetic significance

Godoy Pedro L. 1 pedrolorenagodoy@gmail.com
Bronzati Mario 2 3
Eltink Estevan 4
Marsola Júlio C. de A. 4
Cidade Giovanne M. 4
Langer Max C. 4
Montefeltro Felipe C. 5
1 School of Geography, Earth and Environmental Sciences, University of Birmingham , Birmingham , United Kingdom
2 Bayerische Staatssammlung für Paläontologie und Geologie, Staatlichen Naturwissenschaftlichen Sammlungen Bayerns , Munich , Germany
3 Department of Earth and Environmental Sciences, Palaeontology & Geobiology, Ludwig-Maximilians-Universität München , Munich , Germany
4 Laboratório de Paleontologia de Ribeirão Preto, FFCLRP, Universidade de São Paulo , Ribeirão Preto , Brazil
5 Departamento de Biologia e Zootecnia, Universidade Estadual Paulista (UNESP) , Ilha Solteira , Brazil
Young Mark
Electronic publication date: 2016 May 26
Publication date: 2016
Volume: 4
Electronic Location ID: e2075
Received 2016 Feb 9; Accepted 2016 May 3
Copyright: ©2016 Godoy et al.
Copyright year: 2016
Copyright holder: Godoy et al.
License: This is an open access article distributed under the terms of the Creative Commons Attribution License, which permits unrestricted use, distribution, reproduction and adaptation in any medium and for any purpose provided that it is properly attributed. For attribution, the original author(s), title, publication source (PeerJ) and either DOI or URL of the article must be cited.
License URL: https://creativecommons.org/licenses/by/4.0/

Keywords: Baurusuchidae, Osteoderms, Ecological habit, Adamantina formation, Mesoeucrocodylia, Notosuchia, Phylogenetic bias, Fossil, Paleontology, Bauru group

Funding: University of Birmingham-CAPES Joint Scholarship 3581-14-4 Doris O. and Samuel P. Welles Research Fund Conselho Nacional de Desenvolvimento Científico e Tecnológico (CNPq)—“Ciência sem Fronteiras” Scholarship 246610/2012-3 FAPESP (São Paulo Research Foundation) 2008/57642-6 2009/54656-9 2013/06811-0 2013/23114-1 2013/04516-1 2014/03825-3 Programa de Pós-Graduação em Biologia Comparada, FFCLRP-USP Programa Doutorando com Estágio no Exterior, CAPES 1275/10-0 University of Birmingham Primeiros Projetos PROPe UNESP #730 PLG is supported by a “University of Birmingham-CAPES (Coordenação de Aperfeiçoamento de Pessoal de Nível Superior) Joint Scholarship” (grant number: 3581-14-4), and was also funded by the “Doris O. and Samuel P. Welles Research Fund” to visit the University of California Museum of Paleontology, Berkeley and received additional support from the University of Birmingham. MB is supported by a “Conselho Nacional de Desenvolvimento Científico e Tecnológico (CNPq)—“Ciência sem Fronteiras” Scholarship” (grant number: 246610/2012-3). EE, JCAM, GMC, MCL and FCM were supported by grants from “FAPESP (São Paulo Research Foundation)” (grant numbers: 2008/57642-6; 2009/54656-9; 2013/06811-0; 2013/23114-1; 2013/04516-1; 2014/03825-3) and were funded by the “Programa de Pós-Graduação em Biologia Comparada, FFCLRP-USP”. FCM was also supported by grants from the “Programa Doutorando com Estágio no Exterior, CAPES” (grant number 1275/10-0), “Brazil Visiting Fellows Scheme of the University of Birmingham,” and “Primeiros Projetos PROPe UNESP #730”. The funders had no role in study design, data collection and analysis, decision to publish, or preparation of the manuscript.

==============================
The postcranial anatomy of Crocodyliformes has historically been neglected, as most descriptions are based solely on skulls. Yet, the significance of the postcranium in crocodyliforms evolution is reflected in the great lifestyle diversity exhibited by the group, with members ranging from terrestrial animals to semi-aquatic and fully marine forms. Recently, studies have emphasized the importance of the postcranium. Following this trend, here we present a detailed description of the postcranial elements of Pissarrachampsa sera (Mesoeucrocodylia, Baurusuchidae), from the Adamantina Formation (Bauru Group, Late Cretaceous of Brazil). The preserved elements include dorsal vertebrae, partial forelimb, pelvic girdle, and hindlimbs. Comparisons with the postcranial anatomy of baurusuchids and other crocodyliforms, together with body-size and mass estimates, lead to a better understanding of the paleobiology of Pissarrachampsa sera, including its terrestrial lifestyle and its role as a top predator. Furthermore, the complete absence of osteoderms in P. sera, a condition previously known only in marine crocodyliforms, suggests osteoderms very likely played a minor role in locomotion of baurusuchids, unlike other groups of terrestrial crocodyliforms. Finally, a phylogenetic analysis including the newly recognized postcranial features was carried out, and exploratory analyses were performed to investigate the influence of both cranial and postcranial characters in the phylogeny of Crocodyliformes. Our results suggest that crocodyliform relationships are mainly determined by cranial characters. However, this seems to be a consequence of the great number of missing entries in the data set with only postcranial characters and not of the lack of potential (or synapomorphies) for this kind of data to reflect the evolutionary history of Crocodyliformes.

Introduction

Baurusuchids are important components of the Late Cretaceous crocodyliform faunas (Montefeltro, Larsson & Langer, 2011; Godoy et al., 2014). Despite the uncertainties regarding its relation to Sebecidae, the presence of a monophyletic Baurusuchidae within Notosuchia (Mesoeucrocodylia) is found in many recent analyses (e.g.,: Sereno & Larsson, 2009; Bronzati, Montefeltro & Langer, 2012; Montefeltro et al., 2013; Pol et al., 2014). The group is restricted to South America, with one possible exception in Pakistan (Wilson, Malkani & Gingerich, 2001; Montefeltro, Larsson & Langer, 2011). The group exhibits a peculiar morphology for crocodyliforms, including large size, a dog-like skull with hypertrophied canines and cursorial limb morphology, illustrating their role as top predator in the paleoenvironments they occurred (Montefeltro, Larsson & Langer, 2011; Riff & Kellner, 2011; Godoy et al., 2014).

Most of baurusuchid diversity (8 out of 10) comes from the Bauru Group, in southeast Brazil, including Pissarrachampsa sera, from the Adamantina Formation (Montefeltro, Larsson & Langer, 2011). As typical for descriptive works on crocodyliforms (e.g., Wu, Sues & Sun, 1995; Buckley et al., 2000; Gasparini, Pol & Spalletti, 2006; Novas et al., 2009; O’Connor et al., 2010; Iori & Carvalho, 2011) the original description of Pissarrachampsa sera as exclusively based on its skull morphology. This practice does not seem to be related to the nature of the findings itself, as fossil crocodyliforms are typically found with associated postcranium, as in the case of P. sera. Two partially preserved skulls, including the holotype (Montefeltro, Larsson & Langer, 2011), were collected in 2008. Later expeditions to the type locality, between 2008 and 2010, recovered additional material referred to P. sera, including the postcranial elements described here.

Material and Methods

Systematic paleontology

Crocodyliformes Benton & Clark, 1988	
Mesoeucrocodylia Whetstone & Whybrow, 1983 sensu Benton & Clark, 1988	
Baurusuchidae Price, 1945	
Pissarrachampsinae Montefeltro, Larsson & Langer, 2011	
Pissarrachampsa Montefeltro, Larsson & Langer, 2011	
Pissarrachampsa sera Montefeltro, Larsson & Langer, 2011	

Holotype. LPRP/USP 0019, nearly complete skull and mandibles lacking the cranialmost portion of the rostrum, seven dorsal vertebrae, partial forelimb, pelvic girdle, and hindlimbs.

Previously referred specimens. LPRP/USP 0018, partial rostrum with articulated mandibles.

Additional referred specimens. LPRP/USP 0739, an isolated left pes; LPRP/USP 0740, an isolated right ulna; LPRP/USP 0741, an isolated right tibia; LPRP/USP 0742, an isolated left ilium; LPRP/USP 0743, a partial isolated left femur; LPRP/USP 0744, articulated right femur, tibia and fibula; LPRP/USP 0745, an isolated right manus; LPRP/USP 0746, an isolated right pes.

Type locality. Inhaúmas-Arantes Farm, Gurinhatã (Martinelli & Teixeira, 2015), Minas Gerais state, Brazil (19°20′41.8″S; 49°55′12,9″W). The original description indicated the type locality in the municipality of Campina Verde. However, new information using Global Positioning System (GPS) data show it within the municipality of Gurinhatã.

Age and horizon. Adamantina Formation, Bauru Group, Bauru Basin; Late Cretaceous, Campanian-Maastrichtian (Batezelli, 2015). Note, however, that the stratigraphic nomenclature of the region, as well the ages of the units, is still under debate (see also Fernandes & Coimbra, 1996; Fernandes & Coimbra, 2000; Fernandes, 2004; Batezelli, 2010; Batezelli, 2015; Fernandes & Magalhães Ribeiro, 2014), and the original description of Pissarrachampsa sera (Montefeltro, Larsson & Langer, 2011) considered the type locality as belonging to the Vale do Rio do Peixe Formation.

Appended Diagnosis. Baurusuchid with four maxillary teeth; a longitudinal depression on the rostral portion of frontal; frontal longitudinal ridge extending rostrally overcoming the frontal midlength; supratemporal fenestra with equally developed medial and rostral rims; lacrimal duct at the corner formed by the dorsal and lateral lacrimal surfaces; well developed rounded foramen between the palpebrals; quadratojugal and jugal do not form a continuous ventral border (a notch is present due to the ventral displacement of the quadratojugal); four quadrate fenestrae visible laterally; quadrate lateral depression with rostrocaudally directed major axis; sigmoidal muscle scar in the medial surface of the quadrate; ectopterygoid almost reaching the caudal margin of the pterygoid wings; a single ventral parachoanal fenestra and one ventral parachoanal fossa (divided into medial and lateral parachoanal subfossae); lateral Eustachian foramina larger than the medial one; a deep depression on the caudodorsal surface of the pterygoid wings (Montefeltro, Larsson & Langer, 2011). The following postcranial features were identified as diagnostic for P. sera: ulnar shaft subtriangular in cross-section and strongly bowed laterally; large lateral projection of the supraacetabular crest of the ilium; femur with caudally pointed margin of the medial proximal crest; well-developed femoral “femorotibialis ridge”; short and sharp crest at the craniolateral margin of the distal tibia, ending caudal to the fibular contact of the distal hook; lateral iliofibularis trochanter sharply raised and proximodistally elongated; fibular distal hook contact with tibia placed more proximally relative to the distal articulation of the latter bone; absence of astragalar fossa; restricted anterior hollow on the cranial surface of the astragalus; lateral tubercle at the lateral ridge of calcaneal tuber; complete absence of postcranial osteoderms.

Field work permit

All necessary permits were obtained for the field work, which complied with all relevant regulations. The field work and fossil collection was previously communicated to the Departamento Nacional de Produção Mineral—DNPM, as requested in the ordinance no 4.146 from March 4th, 1942.

Additional information

Nine specimens are described here, including materials associated with the holotype (LPRP/USP 0019), all collected in expeditions to the type locality between 2008 and 2010. The postcranial bones are referred to Pissarrachampsa sera, primarily due to the presence of features compatible with the postcranial morphology of other baurusuchids, but also because the relatively restricted locality “Inhaúmas-Arantes Farm” provided, so far, exclusively materials referred to P. sera. The material assigned to the holotype was not collected at the same time as the skull (Montefeltro, Larsson & Langer, 2011). However, this association is assumed as the postcranial elements were spacially identified during the first expedition placed only few centimeters from the back of the skull, in its natural anatomical position in the same stratigraphic level in the outcrop. Also, it is unlikely that the specimen assignations employed here is wrong, due to discrepant sizes, anatomical overlaps and different locations in the quarry.

Description

The postcranial remains of Pissarrachampsa sera were compared within the context of Crocodyliformes although special attention was given to the morphology of other baurusuchids with postcranium. The comparisons focused on first-hand examination of specimens (Table 1), however, published resources were also used.

Table 1 List of taxa used for comparison in the description.

Taxon	Specimens numbers/references	
Alligator sp.	Brochu (1992)	
Aplestosuchus sordidus	LPRP/USP 0229a	
Araripesuchus gomesii	AMNH 24450; Turner (2006)	
Araripesuchus tsangatsangana	FMNH PR 2297; FMNH PR 2298; FMNH PR 2326; FMNH PR 2327; FMNH PR 2335; FMNH PR 2337; Turner (2006)	
Baurusuchus albertoi	MZSP-PV 140; Nascimento (2008) and Nascimento & Zaher (2010)	
Baurusuchus salgadoensis	UFRJ DG 285-R; Vasconcellos & Carvalho (2010)	
Caiman sp.	LPRP/USP N 0008; MZSP 2137; Brochu (1992) and Nascimento (2008)	
Chimaerasuchus paradoxus	IVPP V8274; Wu & Sues (1996)	
Crocodylus sp.	Brochu (1992)	
Edentosuchus tienshanensis	Pol et al. (2004)	
Lomasuchus palpebrosus	Leardi et al. (2015)	
Mahajangasuchus insignis	FMNH 2721 (research cast of UA8654); Buckley & Brochu (1999)	
Mariliasuchus amarali	UFRJ-DG-50-R, UFRJ-DG-105-R; Nobre & Carvalho (2013)	
Melanosuchus niger	Brochu (1992) and Nascimento (2008)	
Microsuchus schilleri	Leardi, Fiorelli & Gasparini (2015)	
Notosuchus terrestris	MACN-PV RN 1037; MACN-PV RN 1044, MACN-PV N 109; MUCPv-137; Pol (2005) and Fiorelli & Calvo (2008)	
Orthosuchus stormbergii	SAM-PK 409; Nash (1975)	
Protosuchus richardsoni	AMNH 3024; UMCP 34634, 36717	
Sebecus icaeorhinus	AMNH 3159; Pol et al. (2012)	
Sichuanosuchus shuhanensis	Wu et al. (2007)	
Simosuchus clarki	Research cast of UA 8679; Georgi & Krause (2010) and Sertich & Groenke (2010)	
Stratiotosuchus maxhechti	DGM 1477-R; Riff (2007) and Riff & Kellner (2011)	
Theriosuchus pusillus	NHMUK 48330; Wu, Sues & Brinkman (1996)	
Uberabasuchus terrificus	CPPLIP 0630; Vasconcellos (2006)	
Uruguaysuchus aznarezi	Pol et al. (2012)	
Yacarerani boliviensis	Leardi et al. (2015)	

Axial skeleton

Dorsal vertebrae

Seven dorsal vertebrae are partially preserved in the holotype of Pissarrachampsa sera (LPRP/USP 0019), all of which exhibit the typical amphicoelous morphology seen in Notosuchia (Pol, 2005; Nascimento & Zaher, 2010). Five partial vertebrae are articulated in a series, one of which lacks part of the neural arch, (Figs. 1A–1C), and are recognized as mid- to caudal-dorsal vertebrae, whereas the other two are isolated and very likely belong to a more cranial position in the vertebral series (Fig. 1D). One of the features used to determine the axial position of the preserved vertebrae was the relative position of the parapophysis and diapophysis. In notosuchians, as in Baurusuchus albertoi (Nascimento & Zaher, 2010), Sebecus icaeorhinus (Simpson, 1937), and Notosuchus terrestris (Woodward, 1896), the diapophysis is located more dorsally in cranial dorsal vertebrae, but migrate to a more ventral position caudally along the series (Pol, 2005; Nascimento & Zaher, 2010; Pol et al., 2012). On the other hand, the parapophysis is located ventrally in cranial-dorsal vertebrae, and migrate to a more dorsal position in more caudal elements, until it reaches the same dorsoventral level of the diapophysis (Pol, 2005; Nascimento & Zaher, 2010; Pol et al., 2012). The vertebrae in the articulated series show no evidence of para- and diapophyses migration, with both structures located at the same dorsoventral level at the distal portion of the transverse process. In addition, the preserved prezygapophyses are fused with the transverse processes. In closely related taxa, such as Baurusuchus albertoi and Notosuchus terrestris, this fusion is present in vertebrae caudal to the seventh dorsal element (Pol, 2005; Nascimento & Zaher, 2010), also suggesting that this sequence does not belong to cranial-dorsal vertebrae.

Figure 1 Pissarrachampsa sera (holotype, LPRP/USP 0019), photographs and schematic drawing of the articulated dorsal vertebrae in left lateral (A and B) and ventral views (C), and isolated dorsal vertebra in caudal view (D).

Cross-hatched areas represent broken surfaces. Black areas represent sediment-filled areas. Abbreviations: dpon, depression between the postzygapophysis and the neural spine; ns, neural spine (base); ncs, neurocentral suture; pf, postspinal fossa; poz, postzygapophysis; prz, prezygapophysis; tp, transverse process; vc, vertebral centrum. Scale bar equals 5 cm.

The vertebrae of Pissarrachampsa sera have an elliptical centrum in cranial view and are constricted at the middle, as typical for notosuchians (Pol, 2005). The centrum is slightly craniocaudally longer than high (measured from the ventral margin to the level of the ventral limit of the neural channel), and the dimensions are approximately the same in all preserved centra (28 mm long, and 19 mm high). The preserved portion of the neural spine in the third vertebra of the sequence suggests that this structure projects cranially, as in caudal dorsal vertebrae of Baurusuchus albertoi. However, the neural spine of caudal-dorsal vertebrae of Baurusuchus bends caudally on its distal end (Nascimento & Zaher, 2010); a condition not accessible in P sera. The transverse processes are caudally oriented, and project horizontally in cranial and caudal views.

The base of the prezygapophyseal process is located ventral to the upper margin of the neural canal, and projects dorsally and laterally. There is also a slight caudal projection and the prezygapophyses do not extend beyond the cranial limit of the vertebral centrum. The articulation area between the pre- and postzygapophyses is slightly oblique in relation to the horizontal plane of the vertebral column. The postzygapophyses, in the second and third vertebrae of the articulated series, are dorsally curved and project from the caudalmost part of the transverse processes. There is a deep fossa cranial to the postzygapophysis, at the intersection of the neural spine with the transverse process. Pol et al. (2012) suggest that this fossa is exclusively found in notosuchians. The cranial limit of this fossa is marked by a ridge, which extends laterally from the base of the neural spine to half of the lateral length of the transverse process.

One of the isolated vertebrae (Fig. 1D) provides additional information on the vertebral morphology of Pissarrachmpsa sera. The dimensions of this vertebral centrum are approximately the same as for those of the articulated series. However, the neural arch is slightly craniocaudally longer. Also, its neural canal exhibits a rounded opening in cranial view. In caudal view, the postzygapophyses are connected by the postspinal fossa (Pol et al., 2012). The U-shaped ventral margin of this fossa forms a groove located ventral to the dorsal margin of the neural canal (Fig. 1D), a feature that is also observed in cervical and dorsal vertebrae of Baurusuchus albertoi (Nascimento, 2008; Nascimento & Zaher, 2010). This groove becomes progressively wider dorsally, until it merges with the zygapophyses. Also, in dorsal view, the cranialmost part of the fossa is lateromedially narrower than the area between the postzygapophyses.

The suture line between the neural arch and the vertebral centrum is clearly distinguishable in the best preserved isolated vertebra, and it is very likely that the neurocentral suture was also not completely closed in the dorsal vertebrae of the articulated series. Brochu (1996) proposed a cranial to caudal closure pattern of this suture for the crown-group Crocodylia, so that juveniles retain the suture opened in caudal presacral vertebrae. Irmis (2007) observed a similar pattern in phytosaurs and tentatively suggested it is typical of members of the Pseudosuchia lineage, but not of the Avemetatarsalia lineage. However, after analyzing dorsal vertebrae of Notosuchus terrestris, Pol (2005) commented that this pattern described in Brochu (1996) might not be valid for Crocodyliformes outside the Crocodylia clade, such as Pissarrachampsa sera. As the vertebrae described here belong to the holotype, which is likely a mature specimen based on comparisons to smaller specimens from the type locality, our results reinforce the inference of Pol (2005). Finally, Ikejiri (2012) showed that sutures of presacral vertebrae remain opened even in some very mature extant alligators, and Bailleul et al. (2016) have demonstrated that addressing the stage of maturity of archosaurian specimens based on the level of sutural closure in the skull can be misleading. In this context, vertebral sutural closure should not be used as the single factor when inferring the stage of maturity in crocodyliforms.

Figure 2 Pissarrachampsa sera (holotype, LPRP/USP 0019), photographs and schematic drawings of the right ulna in cranial (A and B), lateral (C and D), caudal (E and F), and medial views (G and H).

Light grey represents (broken) articulation areas. Abbreviations: cop, caudal oblique process; cr, caudal ridge; crop, cranial oblique process; crp, ulnar cranial process; ers, M. extensor carpi radialis brevis sulcus; fds, M. flexor digitorum longus insertion surface; fdsc, M. flexor digitorum longus insertion scars; fus, M. flexor ulnaris insertion surface; lp, ulnar lateral process; lr, lateral ridge; olp; olecranon process; pqf; M. pronator quadratus origin fossa; rf, radial facet; tbs, M. triceps brachii insertion scars; vf, vascular foramen. Scale bar equals 5 cm.

Appendicular skeleton

Forelimb

Ulna.

The right ulna of the holotype of Pissarrachampsa sera is preserved (LPRP/USP 0019), as well as a smaller referred right ulna (LPRP/USP 0740) that corresponds to a juvenile individual. The holotypic ulna is damaged at both ends (Fig. 2). Its maximum proximodistal length is 16.5 cm, and the midshaft mediolateral width is 1.8 cm. The general shape is similar to that of other crocodyliform ulnae, including baurusuchids and other notosuchians (Nascimento & Zaher, 2010; Sertich & Groenke, 2010; Vasconcellos & Carvalho, 2010; Riff & Kellner, 2011; Godoy et al., 2014), but less lateromedially compressed than the gracile ulnae of Araripesuchus tsangatsangana (Turner, 2006). The interosseous space between the articulated ulna and radius is reduced, with nearly no space separating the distal and proximal thirds of both bones shafts. Only in the midshaft region this space is noted, although relatively short if compared with the large space seen in extant crocodylians (Brochu, 1992). This pattern is also seen in other terrestrial fossil crocodyliforms, such as baurusuchids Stratiotosuchus maxhechti (Campos et al., 2001) and Baurusuchus albertoi, as well as Araripesuchus tsangatsangana (Turner, 2006; Nascimento & Zaher, 2010; Riff & Kellner, 2011).

The proximal end of the ulna is craniocaudally expanded compared to both the shaft and distal end, as in other crocodyliforms. Since the proximal end is damaged, the structures of the articular surface with the humerus are not preserved. The olecranon process is severely damaged, hampering the assessment of its morphology. Nevertheless, two expansions are preserved in the proximal end, a cranial process and a conspicuous lateral process. Prior to taphonomic damage, the proximal surface of the lateral process corresponded to the ulnar radiohumeral surface, but the radial facet is still preserved. In proximal view, the ulna-radius articulation forms a sinusoidal contact (Fig. 3A). In caudal view, distal to the olecranon processes, scars are seen for the insertion of the M. triceps brachii tendon (Meers, 2003).

The ulnar shaft is subtriangular in cross-section, similar to that of other baurusuchids and Simosuchus clarki (Sertich & Groenke, 2010) (Nascimento & Zaher, 2010; Riff & Kellner, 2011), differing from the ovoid shaft of Araripesuchus tsangatsangana and Mahajangasuchus insignis (Buckley & Brochu, 1999) (Turner, 2006). The shaft is significantly bowed laterally, resembling the flexure seen in Simosuchus clarki and Chimaerasuchus paradoxus (Wu, Sues & Sun, 1995), different from the faint curvature seen in other baurusuchids and extant forms (Brochu, 1992; Wu & Sues, 1996; Nascimento & Zaher, 2010; Sertich & Groenke, 2010; Vasconcellos & Carvalho, 2010; Riff & Kellner, 2011; Godoy et al., 2014). The cranial surface of the shaft bears a vascular foramen proximal to the midheight, close to the medial margin. On the lateral surface (Figs. 2C–2D), distal to the lateral process of the proximal end, there is a groove for the insertion of M. extensor carpi radialis brevis pars ulnaris (Meers, 2003), which is distally delimited by a ridge, caudal to that groove. This ridge also marks the cranial limit of M. flexor ulnaris, which extends distally to the distal condyle (Meers, 2003). As a whole, this lateral ridge extends proximodistally in an almost straight line, and is similar to the marked ridge seen in other baurusuchids, as Stratiotosuchus maxhechti and Baurusuchus albertoi, but more conspicuous than in Araripesuchus tsangatsangana (Nascimento & Zaher, 2010; Riff & Kellner, 2011). On the caudal surface (Figs. 2E–2F), the limit between M. flexor digitorum longus and M. flexor ulnaris is marked by a caudal ridge on the distal portion of the shaft. In Baurusuchus albertoi and Simosuchus clarki this ridge is lesspronounced, giving a more rounded aspect to the caudal surface of the shaft in these taxa (Nascimento, 2008; Sertich & Groenke, 2010). On the medial surface (Figs. 2G–2H), just distal to the proximal end, there is an ovoid fossa for the insertion of M. pronator quadratus (Meers, 2003). It is deeper than in Simosuchus clarki and Araripesuchus tsangatsangana, but does not extend further distally as in Stratiotosuchus maxhechti (Turner, 2006; Sertich & Groenke, 2010; Riff & Kellner, 2011). Due to the fragmentary condition of the region, the flexor ridge that would mark the limit between M. pronator quadratus and M. flexor digitorum longus pars ulnaris (Meers, 2003) is not preserved. However, the latter muscle extends distally until the cranial oblique process of the distal condyle, as seen by the well-marked scars for its insertion proximal to the process, as seen in many fossil taxa (as Baurusuchus albertoi, Stratiotosuchus maxhechti, Simosuchus clarki) and also in living forms (Brochu, 1992; Riff, 2007; Nascimento, 2008; Sertich & Groenke, 2010).

Figure 3 Pissarrachampsa sera (holotype, LPRP/USP 0019), photographs of articulated right ulna and radius in proximal (A) and distal views (B).

Abbreviations: cop, caudal oblique process of ulna; cp, ulnar cranial process; crlp, craniolateral process of ulna; crop, cranial oblique process of ulna; lp, ulnar lateral process; lpc, lateral process of proximal condyle of radius; olp; olecranon process of ulna; rhs, radiohumeral articular surface; rds, radiale articular surface of radius. Scale bar equals 5 cm.

The distal end of the ulna has a craniocaudal breadth 45% shorter than that of the proximal end. The distal condyle has both cranial and caudal oblique processes turned medially. These processes are about the same size, giving the bone a heart-shaped outline in distal view. The craniolateral process is not completely preserved, due to a damage that also affected the distal surface of the condyle, preventing a precise assessment of the ulnare and radiale articulations. Yet, preserved parts suggest the ulnar articulation with the carpal bones was similar to that of other mesoeucrocodylians, such as Stratiotosuchus maxhechti, in which the cranial oblique process articulates with the radiale and the caudal process articulates with the ulnare (Riff & Kellner, 2011).

Figure 4 Pissarrachampsa sera (holotype, LPRP/USP 0019), photographs and schematic drawings of the right radius in cranial (A and B), lateral (C and D), caudal (E and F), and medial views (G and H).

Light grey represents articulation areas. Abbreviations: ars, M. abductor radialis insertion surface; bbt, M. biceps brachii insertion tubercle; has, M. humeroantebrachialis inferior insertion scar; ecrs, M. extensor carpi radialis brevis insertion surface; hrt, M. humeroradialis insertion tubercle; lcr, thin longitudinal crest; lpc, lateral process of proximal condyle; mpc, medial process of proximal condyle; pmr, proximodistal medial ridge; pqs, M. pronator quadratus insertion surface; pts, M. pronator teres insertion surface; rds, radiale articular surface; rhs, radiohumeral articular surface; sps, M. supinator insertion surface; uac, ulnar articulation concavity; uf, ulnar facet; vf, vascular foramen. Scale bar equals 5 cm.

Radius.

The right radius is preserved in the holotype of Pissarrachampsa sera (LPRP/USP 0019). The straight proximodistal extension of its slender shaft gives the bone a rod-like shape; which seems to be exaggerated due to the badly preserved proximal and distal ends (Fig. 4). Its maximum proximodistal length is 16 cm, and the midshaft mediolateral width is 1.4 cm. This general shape resembles that of other baurusuchid radii (Nascimento & Zaher, 2010; Vasconcellos & Carvalho, 2010; Godoy et al., 2014), but it is less robust than in Stratiotosuchus maxhechti (Riff & Kellner, 2011) and in extant crocodylians, such as Caiman and Alligator (Brochu, 1992).

The lateral and medial processes of the proximal condyle are not complete but the lateromedial expansion of the proximal end is clear, as in most crocodyliforms (Pol, 2005). The proximal end of the radius is bent cranially at an angle of approximately 25°. In cranial view (Figs. 4A–4B), the radiohumeral articular surface bears a concavity for the articulation of the radial condyle of the humerus. In caudal view (Figs. 4E–4F), part of a crest is seen, adjacent to the lateral process of the proximal condyle. This crest is described by Pol (2005) for Notosuchus terrestris as a thin proximodistal crest and is also present in Simosuchus clarki, as well as in the baurusuchids Stratiotosuchus maxhechti and Baurusuchus albertoi (Nascimento & Zaher, 2010; Sertich & Groenke, 2010; Riff & Kellner, 2011). The ulnar facet is poorly preserved, but it is represented in caudal view by a concavity between the lateral and medial processes. The medial process of the proximal condyle bears, on its medial surface, the scar for the tendon of M. humeroantebrachialis inferior (Figs. 4E–4H). This scar was described by Turner (2006) for Araripesuchus tsangatsangana, and is also present in Simosuchus clarki and Baurusuchus albertoi (Nascimento & Zaher, 2010; Sertich & Groenke, 2010). Caudodistal to this scar, the tubercle for the insertion of M. biceps brachii is seen (Meers, 2003).

The radial shaft is elliptical in cross-section, and marked by scars and ridges for muscle insertions. In cranial view (Figs. 4A–4B), distal to the proximal condyle, the scar for the M. abductor radialis insertion is present, lateral to the tuberosity for the insertion of M. humeroradialis. This scar extends distally to the midlenght of the shaft, as in other notosuchians and living crocodylians (Meers, 2003; Pol, 2005; Turner, 2006; Sertich & Groenke, 2010). More distally, in the midline of the cranial surface, a proximodistally elongated ridge separates the insertions of M. supinator laterally and M. pronator teres, medially, along most of the shaft (Meers, 2003). This ridge is also seen in Baurusuchus albertoi, but less marked than in Stratiotosuchus maxhechti (Nascimento & Zaher, 2010; Riff & Kellner, 2011). The proximodistally long insertions of M. extensor carpi radialis brevis and M. pronator quadratus are better seen, respectively, on the lateral and caudal surfaces (Figs. 4C–4F) (Meers, 2003). A well-developed, proximodistal elongated ridge marks the caudal limit of M. extensor carpi radialis brevis and the lateral limit of M. pronator quadratus (Meers, 2003) at the lateral surface of the distal half of the shaft (Figs. 4C–4D). This ridge extends from the first to the third quarters of the shaft, resembling that of Simosuchus clarki, Baurusuchus albertoi and Aplestosuchus sordidus (Godoy et al., 2014) (Sertich & Groenke, 2010; Nascimento & Zaher, 2010), but is smoother than that of Stratiotosuchus maxhechti (Riff & Kellner, 2011). Also in lateral view, another ridge, in the proximal half of the shaft, separates the insertion extensions of M. extensor carpi radialis brevis and M. abductor radialis (Meers, 2003). This ridge almost reaches the cranial surface, as in other baurusuchids, differing from the pattern seen in Simosuchus clarki, in which the ridge is restricted to the lateral surface (Sertich & Groenke, 2010; Nascimento & Zaher, 2010; Riff & Kellner, 2011; Godoy et al., 2014).

The distal end of the radius is lateromedially expanded and strongly compressed craniocaudally. In distal view, the caudal surface is concave for the articulation with the ulna (Fig. 3B). On the caudal surface of the distal end (Figs. 4E–4F) a small vascular foramen is seen medial to the ulnar articulation concavity. The radiale articulates with the cranial convex surface of the radius. This articulation gives the radial distal end two separate condyles, a more distally extended medial condyle and a lateral one, as seen in Stratiotosuchus maxhechti and Simosuchus clarki (Sertich & Groenke; Riff & Kellner, 2011).

Carpus.

The holotype (LPRP/USP 0019) has both right radiale and ulnare preserved, along with an incomplete right manus (Fig. 5). Only the cranial surfaces of both bones are visible. The pisiform and the distal carpal, which complete the carpus of Crocodylia, are not preserved in Pissarrachampsa sera (Mook, 1921; Nascimento & Zaher, 2010; Sertich & Groenke, 2010). Both radiale and ulnare are elongated bones, a synapomorphy of Crocodylomorpha (Walker, 1970; Clark, 1986; Benton & Clark, 1988). They are very constricted lateromedially and craniocaudally compressed between enlarged proximal and distal ends, as in Simosuchus clarki, Stratiotosuchus maxhechti and Baurusuchus albertoi (Riff, 2007; Nascimento & Zaher, 2010; Sertich & Groenke, 2010), differently from the highly elongated and slender carpals of Araripesuchus tsangatsangana (Turner, 2006).

The proximal surface of the right radiale of Pissarrachampsa sera (holotype, LPRP/USP 0019) is not completely exposed. However, as the preserved medial two-thirds of the surface are concave, this appears to be also the condition of the lacking portion, whereas the lateral third is occupied by a proximally directed convex lateral process. The same pattern is found in Simosuchus clarki, Stratiotosuchus maxhechti, Notosuchus terrestris, Baurusuchus albertoi, Sebecus icaeorhinus, and Yacarerani boliviensis (Novas et al., 2009) (Pol, 2005; Riff, 2007; Nascimento & Zaher, 2010; Sertich & Groenke, 2010; Pol et al., 2012; Leardi et al., 2015). The exposed portion of the proximal surface represents the articulation for the distal end of the radius, as described for Baurusuchus albertoi, Simosuchus clarki, Stratiotosuchus maxhechti and Araripesuchus tsangatsangana (Turner, 2006; Riff, 2007; Nascimento & Zaher, 2010; Sertich & Groenke, 2010). The presence of a marked longitudinal crest in the cranial surface of the radiale has been described for several notosuchians, such as Notosuchus terrestris, Baurusuchus albertoi, Sebecus icaeorhinus, Stratiotosuchus maxhechti, and Yacarerani boliviensis (Pol, 2005; Riff, 2007; Nascimento & Zaher, 2010; Sertich & Groenke, 2010; Pol et al., 2012; Leardi et al., 2015). On the other hand, Turner (2006) describes a “median ridge” in Araripesuchus tsangatsangana, which may correspond to the longitudinal crest. There is no sign of such a crest in the exposed surface of the radiale of Pissarrachampsa sera, but its absence cannot be confirmed as most of the cranial surface of the radiale is embedded in the rock matrix.

Figure 5 Pissarrachampsa sera (holotype, LPRP/USP 0019), photographs of the right carpus/manus in dorsal (A) and ventral views (B).

Abbreviations: I mc, metacarpal I; II mc, metacarpal II; III mc, metacarpal III; IV mc, metacarpal IV; V mc, metacarpal V; dph, distal phalanx; mph, medial phalanx; pph, proximal phalanx; rdl, radiale; uln, ulnare. Scale bar equals 5 cm.

Sertich & Groenke (2010) described a prominent pit and a raised rugosity for Simosuchus clarki, which topologically corresponds to the proximal portion of the cranial longitudinal crest in Mahajangasuchus insignis, and represents the insertion of the M. extensor carpi radialis longus (Meers, 2003). The presence of raised scars medial and lateral to this pit is has also been described for Simosuchus clarki, consistent with the origin of the superficial extensor muscles for digits I, II and III (Brochu, 1992; Meers, 2003; Sertich & Groenke, 2010). In Pissarrachampsa sera, despite the lack of the pit, it is possible that the exposed surface of the radiale includes the insertion areas of those extensor muscles, or at least those lateral to the pit in Simosuchus clarki.

The ulnare of Pissarrachampsa sera (holotype, LPRP/USP 0019) seems to be proximodistally shorter than the radiale (Fig. 5), as in Araripesuchus tsangatsangana, Baurusuchus albertoi, Simosuchus clarki, Stratiotosuchus maxhechti, Notosuchus terrestris, Yacarerani boliviensis, and Crocodylia (Mook, 1921; Pol, 2005; Turner, 2006; Nascimento & Zaher, 2010; Sertich & Groenke, 2010; Leardi et al., 2015). Its proximal articular surface is covered by matrix, but its proximal outline seems to be subtriangular, with the apex positioned cranially, as in Simosuchus clarki (Sertich & Groenke, 2010).

The distal end of the ulnare is more expanded than the proximal, as in Notosuchus terrestris, Sichuanosuchus shuhanensis (Wu, Sues & Dong, 1997), Baurusuchus albertoi, Araripesuchus tsangatsangana, Stratiotosuchus maxhechti, Simosuchus clarki, Yacarerani boliviensis, and most non-Crocodylia crocodyliforms (Pol, 2005; Turner, 2006; Riff, 2007; Nascimento & Zaher, 2010; Sertich & Groenke, 2010; Leardi et al., 2015). Yet, the bone is not exposed enough to see if this expansion is symmetrical, as in Simosuchus clarki and Yacarerani boliviensis, or more marked medially, as in Notosuchus terrestris, Stratiotosuchus maxhechti and Baurusuchus albertoi (Leardi et al., 2015)

Manus.

Two right manus are associated to Pissarrachampsa sera, one of the holotype (LPRP/USP 0019) and an isolated one (LPRP/USP 0745). The holotypic right manus (Fig. 5) is composed of five digits: the first includes the metacarpal and the proximal phalanx; the second includes the metacarpal, a poorly preserved proximal phalanx, and the distal phalanx; the third includes the metacarpal and fragments of the medial portions of three phalanges; the last two digits include only the metacarpals. The right manus of LPRP/USP 0745 preserves (albeit partially) all five metacarpals, an incomplete proximal phalanx of the digit I, and a fragment that might represent the proximal phalanx of the digit III. The holotypic manus is better seen in ventral view (Fig. 5B), whereas LPRP/USP 0745 has only its dorsal surface exposed.

From the first to the fourth digits, the metacarpals show a decrease in width and an increase in length (Fig. 5B), as in Baurusuchus albertoi and Stratiotosuchus maxhetchi (Nascimento & Zaher, 2010; Riff & Kellner, 2011). Metacarpal I is the most robust, as in Notosuchus terrestris, Stratiotosuchus maxhechti, Simosuchus clarki, and Yacarerani boliviensis, differing from Crocodylia, in which metacarpal I is similar in robustness to the others (Mook, 1921; Pol, 2005; Sertich & Groenke, 2010; Riff & Kellner, 2011; Leardi et al., 2015). The preserved proximal end of metacarpal V is dorsoventrally flat and lateromedially wide, as in Baurusuchus albertoi, S. maxhetchi, and Yacarerani boliviensis (Nascimento & Zaher, 2010; Riff & Kellner, 2011; Leardi et al., 2015).

All phalanges preserved in the holotype are robust, with a blocky appearance in dorsal and ventral views, with a midlength constriction, also seen in Baurusuchus albertoi, Simosuchus clarki, Stratiotosuchus maxhetchi, Araripesuchus tsangatsangana, and Yacarerani boliviensis (Turner, 2006; Nascimento & Zaher, 2010; Sertich & Groenke, 2010; Riff & Kellner, 2011; Leardi et al., 2015). All manual phalanges of Pissarrachampsa sera that preserve their articular surfaces exhibit medial and lateral condyles, in both the distal and proximal surfaces.

Pelvic girdle

Ilium.

One left ilium is partially preserved for Pissarrachampsa sera (Fig. 6), from a referred specimen (LPRP/USP 0742). It lacks the distal part of the postacetabular process, most of the preacetabular process, and the ventral portion of the acetabular region. The acetabulum is deep, as in Baurusuchus albertoi and Sebecus icaeorhinus, as a result from the strictly lateral orientation of the supraacetabular crest (Nascimento & Zaher, 2010; Pol et al., 2012). On the other hand, the supraacetabular crest of Araripesuchus tsangatsangana projects not only laterally, but also dorsally, which gives a shallower aspect to the acetabulum (Turner, 2006). In some neosuchians and living taxa, the crest is strongly inclined dorsally, giving an accentuated shallow aspect to the acetabulum in lateral view (Leardi, Fiorelli & Gasparini, 2015).

Figure 6 Pissarrachampsa sera (LPRP/USP 0742), photographs and schematic drawing of the left ilium in dorsal (A and B), medial (C and D), and lateral views (E).

Cross-hatched areas represent broken surfaces. Abbreviations: ac, acetabulum; acr, acetabular roof; das, dorsal portion of the articular surface for the second sacral rib; dmar, dorsal margin of the acetabular roof; pap, postacetabular process; imr, ridge on the medial surface of the ilium; s 1r, articular surface for first sacral rib; s 2r, articular surface for second sacral rib. Scale bar equals 5 cm.

In Pissarrachampsa sera, the morphology of the dorsal surface of the acetabular roof resembles that of Baurusuchus albertoi (Figs. 6A–6B) (Nascimento & Zaher, 2010). In both taxa, the dorsal component of the supraacetabular crest is confluent with the remaining dorsal portion of the bone, extending as a flat horizontal surface, giving the ilium a broad aspect. On the other hand, in Sebecus icaeorhinus, Microsuchus schilleri (Dolgopol de Sáez, 1928), and living forms, such as Caiman latirostris (Daudin, 1802) (MZSP 2137), the supraacetabular crest is not confluent with the rest of the dorsal margin, but has a medial boundary (Pol et al., 2012; Leardi, Fiorelli & Gasparini, 2015). In Sebecus icaeorhinus and Caiman yacare (Daudin, 1802), the dorsal margin is sloped, with the portion corresponding to the supraacetabular crest lying dorsal to the medial portion of the iliac dorsal surface (Nascimento, 2008; Pol et al., 2012). Given the great lateral projection of the supraacetabular crest, the maximum width of the dorsal margin of the ilium of Pissarrachampsa sera is located right above the caudal margin of the acetabular area. The rest of the dorsal surface becomes gradually narrower in the direction of both the pre- and postacetabular processes. Rugosities on the dorsal surface of the supraacetabular crest indicate the area for the attachment of M. iliotibialis 1 and 2 (Romer, 1923; Leardi, Fiorelli & Gasparini, 2015). In Pissarrachampsa sera, most of this surface is rugose, indicating a greater area for the attachment of those muscles.

The proximal portion of the postacetabular process is at least four times dorsoventrally higher than lateromedially wide, and its dorsal margin is slightly caudoventrally oriented. In medial view, it is possible to see the medial expansion of the dorsal portion of the postacetabular process, forming a ridge that extends craniocaudally (Figs. 6C–6D). This ridge marks the dorsal limit of a concave surface on the medial portion of the ilium. Ventrally, this concavity is delimited by a curved ridge, which corresponds to the dorsal part of the articular surface for the second sacral rib (see Pol et al., 2012), and this same morphology is also seen in Baurusuchus albertoi and Sebecus icaeorhinus (Nascimento & Zaher, 2010; Pol et al., 2012). On the other hand, in Theriosuchus pusillus (Owen, 1879) and some extant taxa, such as Caiman yacare and Melanosuchus niger (Spix, 1825), there is no evidence of a supraacetabular process medial crest, which gives a more flattened aspect to the process above the articular surface for the second sacral rib (Wu, Sues & Brinkman, 1996). Baurusuchus albertoi has a total of three sacral vertebrae, with the articulation surface for the third element located in the distal portion of the postacetabular process (Nascimento & Zaher, 2010). Three sacral vertebrae are also found in other baurusuchids, such as Baurusuchus salgadoensis (Carvalho, Campos & Nobre, 2005) (Vasconcellos & Carvalho, 2010) and Aplestosuchus sordidus (Godoy et al., 2014), and there is no evidence of a different condition in Pissarrachampsa sera, although this remains speculative due to the absence of more complete remains.

Ischium.

Both left and right ischia of the holotype of Pissarrachampsa sera (LPRP/USP 0019) are partially preserved, lacking the distal portions of the ischial blade, and of the iliac and pubic peduncles. Despite the incompleteness, the typical crocodyliform ischial morphology is recognizable (Figs. 7A–7B), with a lateromedially constricted ischial blade, a caudal process which would probably contact the ilium, and a cranial process which likely contacted both ilium and pubis (Sertich & Groenke, 2010). The notch between both processes formed the ventral margin of the perforate acetabulum, similar to the condition seen in mesoeucrocodylians such as Chimaerasuchus paradoxus, Mahajangasuchus insignis, Stratiotosuchus maxhechti, and Sebecus icaeorhinus (Wu & Sues, 1996; Buckley & Brochu, 1999; Riff & Kellner, 2011; Pol et al., 2012). The proximal parts of both processes differ in thickness, with a more extended cranial process, as seen in Stratiotosuchus maxhechti and Sebecus icaeorhinus (Riff & Kellner, 2011; Pol et al., 2012). In these two taxa, however, the cranial process expands distally, becoming more robust, an unknown condition for Pissarrachampsa sera.

Figure 7 Pissarrachampsa sera (holotype, LPRP/USP 0019), photographs and schematic drawing of left ischium in lateral view (A and B) and pubis in caudal view (C).

Abbreviations: ac, acetabulum; ib, iliac blade; ipi, iliac peduncle of ischium; ph, pubic head; ps, pubic symphysis; psh, pubic shaft; ppi, pubic peduncle of ischium; ri, ridge; ti, tubercle of the ischium. Scale bar equals 5 cm.

On the lateral surface of the ischial blade (Figs. 7A–7B), a ridge extends dorsoventrally along its proximal third marking the limits of muscles attached to the ischium. The ischium is very constricted lateromedially, cranial and caudal to this ridge, giving a sharp aspect to its margins. Caudal to the ridge is the area for attachment of both M. flexor tibialis internus pars 3 laterally and M. ischiotrochantericus medially (Hutchinson, 2001a). In the distal portion of the ischial blade, only the cranial margin is constricted, as the dorsoventral ridge becomes confluent with the caudal margin, which becomes more rounded. The constricted cranial margin corresponds to the attachment surface for M. puboischiofemoralis externus pars 3, on the medial surface of the bone (Hutchinson, 2001a; Riff, 2007). In cranial and lateral views, it is possible to see a tubercle on the dorsal portion of the ischial blade, ventral to the cranial process of the ischium. Stratiotosuchus maxhechti bears a similar tubercle, which is interpreted as the attachment point for M. pubioischiotibialis (Riff & Kellner, 2011).

Pubis.

Both pubes are partially preserved (Fig. 7C) in the holotype of Pissarrachampsa sera (LPRP/USP 0019). As is typical for Crocodyliformes, the proximal shaft of the pubis lacks the obturator foramen present in some non-Crocodyliformes Crocodylomorpha, such as Terrestrisuchus gracilis (Crush, 1984). In general, the pubis has a rod-like aspect, as also seen in Baurusuchus albertoi, Stratiotosuchus maxhechti and the protosuchians Protosuchus richardsoni (Brown, 1933) and Orthosuchus stormbergii (Nash, 1968) (Colbert & Mook, 1951; Nash, 1975; Nascimento & Zaher, 2010; Pol et al., 2012). On the other hand, other crocodyliforms such as Araripesuchus tsangatsangana, Notosuchus terrestris, Mahajangasuchus insignis, Theriosuchus pusillus, as well as the living forms, bear an expanded distal pubic end (Brochu, 1992; Wu, Sues & Brinkman, 1996; Buckley & Brochu, 1999; Turner, 2006; Pol, 2005).

Given the incompleteness of the pelvis of Pissarrachampsa sera, the isolation of the pubis from the acetabulum cannot be asserted. Yet, in all Crocodyliformes, except protosuchians, the pubis is excluded from the acetabulum by the cranial process of the ischium, which represents the articulation point for the proximal end of the pubis (Colbert & Mook, 1951). In Pissarrachampsa sera, the partially preserved proximal articulation is lateromedially constricted, and more constricted in its cranial third, giving it a pear-shaped aspect. This lateromedial constriction extends distally along the shaft, as also seen in Stratiotosuchus maxhechti (Riff, 2007). Pissarrachampsa sera and Stratiotosuchus maxhechti also share the proximal pubic shaft bent approximately 30 degrees in relation to the pubic blade. In other notosuchians, such as Araripesuchus tsangatsangana and Simosuchus clarki, and also in the living Crocodylia, such bending is unknown (Turner, 2006; Riff, 2007; Sertich & Groenke, 2010). The pubic blade is craniocaudally constricted in its medial third, which forms the pubic symphysis. Lateral to the laminar symphyseal region, the ischial blade does not show any evidence of the craniocaudal constriction. The attachment area for both M. puboischiofemoralis externus pars 1 and 2 is probably located in the proximal two thirds of the transitional area between the constricted and non-constricted regions of the pubic blade, in the caudal and cranial surfaces respectively (Romer, 1923).

The pubis is a remarkably long element in Pissarrachampsa sera when compared to that of other crocodyliforms. Indeed, even without the distalmost part, the pubic length of Pissarrachampsa sera is 70% of the total length of the femur. This condition is “more similar to that of Stratiotosuchus maxhechti (Riff, 2007), in which this ratio is 80%, than to the condition observed in other crocodyliforms: 25% in Araripesuchus tsangatsangana; 42% in Edentosuchus tienshanensis (Young, 1973); 55% in Sunosuchus junggarensis (Young, 1948); 55% in Mahajangasuchus insignis, and 57% in Caiman yacare (Buckley & Brochu, 1999; Pol et al., 2004; Turner, 2006).

Hindlimb

Femur.

There are four preserved femora known for Pissarrachampsa sera. The femoral pair of the holotype (LPRP/USP 0019), as well as two smaller isolated and partially preserved left and right elements (LPRP/USP 0743 and LPRP/USP 0744). The smaller right femur is still in articulation with tibia and fibula, but the following description is based mostly on the holotypic material (Fig. 8), since these are better preserved. The femur is virtually straight in cranial and caudal views, and its proximodistal length is about 24 cm. It is longer than the tibia and or fibula, as seen in most other Mesoeucrocodylia (Leardi, Fiorelli & Gasparini, 2015). In medial and lateral views, the shaft is slightly bowed cranially, and the proximal and distal ends are cranially and caudally curved. The proximal articulation surface is medially inturned, as seen in Baurusuchus albertoi and Stratiotosuchus maxhechti, but not as displaced as in Araripesuchus tsangatsangana (Turner, 2006; Nascimento & Zaher, 2010; Riff & Kellner, 2011). In proximal view (Figs. 8I–8J), the robust articular surface is rounded and rugose at its distal portion, with scars for muscle insertion, whereas the caudolateral extension of the head is slender, as in other baurusuchids and Mariliasuchus amarali (Carvalho & Bertini, 1999) (Nascimento & Zaher, 2010; Riff & Kellner, 2011; Nobre & Carvalho, 2013). At this point, in caudal view (Figs. 8E–8F), there is a proximodistally extensive “greater trochanter” placed laterally, extending cranially and parallel to the “medial proximal crest,” at the caudal most extension of the head (Pol et al., 2012). The “medial proximal crest” turns caudally in Pissarrachampsa sera, and not medially as in Sebecus icaeorhinus (Pol et al., 2012).

Figure 8 Pissarrachampsa sera (holotype, LPRP/USP 0019), photographs and schematic drawings of the left femur in cranial (A and B), medial (C and D), caudal (E and F), lateral (G and H), proximal (I and J), and distal views (K and L).

Areas of musculature insertion are shadowed in dark gray. Light grey represents areas of bone articulation. Abbreviations: af?, adductor fossa; add1 + 2, M. adductor femoris 1 & 2; cfb, M. caudofemoralis brevis; cfl, M. caudofemoralis longus; crf, cranial flange; fmte, M. femorotibialis externus; fmti, M. femorotibialis internus; ftr, femorotibialis ridge; ga, M. gastrocnemius; gt, greater trochanter; if, M. iliofemoralis; icf, intercondylar fossa; it, M. ischiotrochantericus; lc, lateral condyle; lic, linea intermuscularis caudalis; mc, medial condyle; mpc, medial proximal crest; mscr, medial supracondylar crest; pas, proximal articulation surface; pf, popliteal fossa ; pife, M. puboischiofemoralis externus; pifi 1, M. puboischiofemoralis internus 1; pifi 2, M. puboischiofemoralis internus 2; s fi, articular surface for fibula; smi, surface for muscular insertion; vf, vascular foramen; 4th, fourth trochanter. Scale bar equal 5 cm (A–H) and 2 cm (I–M).

In lateral view (Figs. 8G–8H), the proximal part of the femur bears marked depressions and scars for musculature insertion. The scars along the “greater trochanter” correspond to the insertions of M. ischiotrochantericus and M. puboischiofemoralis internus 2, and are also possibly related to the adductor fossa, placed cranially to these muscles insertions (Hutchinson, 2001b; Sertich & Groenke, 2010; Nascimento & Zaher, 2010). In caudal view (Figs. 8E–8F), M. puboischiofemoralis externus (Hutchinson, 2001b) attaches at the “medial proximal crest.” In cranial view (Figs. 8A–8B), the “cranial flange” marks the transition between the proximal femur and the shaft. There are many names for this structure in the literature: anteromedial process (Fiorelli & Calvo, 2007), anterior flange and caudofemoralis flange (Turner, 2006), and cranium-medial crest (Riff, 2007; Nascimento & Zaher, 2010). Although less sharp and prominent than in Simosuchus clarki, this structure is well marked, and bears scars for musculature insertions (Sertich & Groenke, 2010). This condition is similar to that of other baurusuchids and Araripesuchus tsangatsangana, but Microsuchus schilleri and other small notosuchians, such as Mariliasuchus amarali, have a less marked “cranial flange,” which is absent in Sebecus icaeorhinus and Yacarerani boliviensis (Nobre & Carvalho, 2006; Turner, 2006; Nascimento & Zaher, 2010; Riff & Kellner, 2011; Pol et al., 2012; Nobre & Carvalho, 2013; Leardi et al., 2015). In Pissarrachampsa sera, the “cranial flange” divides the femoral shaft in medial and lateral parts. In cranial view (Figs. 8A–8B), the insertion for M. puboischiofemoralis internus 1 is flanked medially by a rugose convexity related to M. caudofemoralis longus (Hutchinson, 2001b). Caudal to that, another smaller rough convexity, also seen in Araripesuchus tsangatsangana, may correspond to the fourth trochanter (Turner, 2006). This corresponds to a shallow proximodistally oriented groove that extends distally as a faint ridge and has scars for the insertion of M. caudofemoralis brevis (Hutchinson, 2001b). It differs from the poorly developed fourth trochanter of Sebecus icaeorhinus, Microsuchus schilleri, and Yacarerani boliviensis and the very prominent structure seen in Simosuchus clarki (Sertich & Groenke, 2010; Pol et al., 2012; Leardi, Fiorelli & Gasparini, 2015; Leardi et al., 2015).

Other muscle scars seen along the shaft, as well as a foramen mediodistal to the cranial flange. Laterodistal to the flange lies the insertion area for the M. iliofemoralis (Hutchinson, 2001b) and distal to the flange, there is an extensive intermuscular line that almost reaches the proximal limit of the intercondylar fossa (Romer, 1956). This corresponds to the M. femorotibialis internus (Hutchinson, 2001b) and its distal most extension forms a longitudinal ridge, named here “femorotibialis ridge.” This intermuscular line does not form a ridge in the juvenile specimen, and is interpreted as an ontogeny-related character. Caiman sp. (LPRP/USP N 0008) also has this intermuscular line, but it does not form a ridge. The presence of this ridge is not clear in other notosuchians, except for Stratiotosuchus maxhecthi and Aplestosuchus sordidus, in which it is smoother than in Pissarrachampsa sera (Riff & Kellner, 2011; Godoy et al., 2014). On the caudal face of the femoral shaft (Figs. 8E–8F), the linea intermuscularis caudalis extends obliquely, from the fourth trochanter to the proximal portion of the lateral condyle, and forms the lateral border of the popliteal fossa. This scar corresponds to the boundary between M. femorotibialis externus, craniomedially, and M. adductor femoris 1 & 2, caudolaterally (Hutchinson, 2001b).

The two distal condyles are well developed, forming the intercondylar fossa cranially and a deep popliteal fossa caudally. The latter is rugose, as in Stratiotosuchus maxhechti, whereas the intercondylar fossa has smoother scars for muscle insertions (Romer, 1956; Riff & Kellner, 2011). The lateral or fibular condyle has a laterodistal concavity, possibly related to the fibular articulation. It is about two times larger than the medial or tibial condyle, which is not as distally expanded as the lateral condyle, a general crocodyliform condition (Sertich & Groenke, 2010; Pol et al., 2012). In lateral view (Figs. 8G–8H), the rugose surface above the lateral condyle makes the insertion of M. gastrocnemius (Brochu, 1992; Sertich & Groenke, 2010). Cranially, the distal portion of the femur has a well developed medial supracondylar ridge, whereas the lateral supracondylar ridge is smoother. This differs from the condition in Sebecus icaeorhinus, which lacks a marked transition from the cranial to the lateral surfaces of the distal femur (Pol et al., 2012). The caudal surface (Figs. 8E–8F) of the distal femur bears both medial and lateral supracondylar ridges (the latter would be the distal extension of the linea intermuscularis caudalis), as well as a popliteal fossa between these (Hutchinson, 2001b; Pol et al., 2012). The medial supracondylar ridge forms a proximodistally oriented crest, above the medial condyle, separating the caudal and lateral surfaces of the distal portions of the femur. The medial facet of the distal portion of the femur is almost flat, cranially bound by the medial supracondylar ridge, whereas in Sebecus icaeorhinus this surface is slightly convex (Pol et al., 2012).

Tibia.

Both tibiae of the holotype (LPRP/USP 0019) are nearly complete, and articulated with the fibulae in their original position (Fig. 9). Additionally, there is a smaller isolated right tibia (LPRP/USP 0741), as well as the additional right tibia in articulation with femur and fibula (LPRP/USP 0744). The shafts of the articulated tibia and fibula are very close to one another (Figs. 9A–9B), as are the radius and ulna. This condition is different from that of modern crocodylians (e.g., Caiman and Melanosuchus) in which this distance is larger. The tibia of Pissarrachampsa sera is similiar in robustness to the tibiae of most crocodyliforms, differing from the more gracile elements of Araripesuchus tsangatsangana and Microsuchus schilleri (Brochu, 1992; Turner, 2006; Leardi, Fiorelli & Gasparini, 2015). The tibia is 18.6 cm long, i.e., 77% the femur’s length, the same ratio of Sebecus icaeorhinus. This differs from other notosuchians, such as the relatively short tibia of other baurusuchids, such as Baurusuchus albertoi and Stratiotosuchus maxhechti, (about 72%) and the elongated bone (82%) of Araripesuchus tsangatsangana (Pol et al., 2012).

Figure 9 Pissarrachampsa sera (holotype, LPRP/USP 0019), photographs and schematic drawings of the articulated left tibia and fibula in caudal (A and B), lateral (C and D), cranial (E and F), and medial views (G and H).

Light grey represents areas of bone articulation. Arrow indicates a “sharp crest.” Abbreviations: dh, distal hook; ffx, fossa flexoria; ift, iliofibularis trochanter; ill, internal lateral ligament; lell, long external lateral ligament; lf, lateral facet; mf, medial facet; mfdl, origin of M. flexor digitorium longus; mfti, M. flexor tibialis internus insertion; mic, M. interosseous cruris insertion; mta, M. tibialis anterior insertion; vf, vascular foramen. Scale bar equals 5 cm.

The proximal and distal extremities of the tibia are mediolaterally well expanded. The proximal surface is divided into medial and lateral facets (Figs. 9A–9B), which respectively correspond to the articulation areas for the tibial and fibular condyles of the femur. In proximal view, the medial articulation (posteromedial proximal process of the tibia, according to Leardi et al., 2015) has a trapezoid-shape; a pattern also seen in other baurusuchids, such as Stratiotosuchus maxhechti and Baurusuchus albertoi (Nascimento & Zaher, 2010; Riff & Kellner, 2011). The medial articular facet is more protruded relative to the lateral one. The proximal surface of the medial facet forms a gentle concavity, corresponding to the “proximal pit” sensu Brochu (1992), and bears a pronounced deflection toward its caudomedial corner (Fig. 9). This condition is also observed in Sebecus icaeorhinus, which bears a gently protruded medial facet, but differs from Mariliasuchus amarali, Yacarerani boliviensis, and Stratiotosuchus maxhechti, in which that medial portion is weakly pronounced (Pol et al., 2012; Leardi et al., 2015). The latter condition is also present in modern crocodylians (e.g., Caiman, Melanosuchus and Alligator) resulting in equally projected facets. The lateral articular facet is semi-lunar in shape and slightly concave in proximal view. The cranial border is rounded and the caudal tip is somewhat deflected distally. It resembles the pattern of Sebecus icaeorhinus and Yacarerani boliviensis, differing from the weakly projected tip of Mariliasuchus amarali, Araripesuchus tsangatsangana and Stratiotosuchus maxhechti (Turner, 2006; Riff & Kellner, 2011; Pol et al., 2012; Nobre & Carvalho, 2013; Leardi et al., 2015).

Cranially, the proximal expansion of the tibia bears a well-developed tuberosity for the insertion of M. flexor tibialis internus (Figs. 9E–9F). This insertion is proximodistally elongated, as in Araripesuchus tsangatsangana, but it is more sharply raised and closer to the proximal articular surface, a condition more marked than in extant taxa (e.g., Alligator, Caiman and Melanosuchus). Proximolaterally, there is a shallow depression related to the attachment of the internal lateral ligament (Figs. 9E–9F), as in Alligator mississippiensis (Daudin, 1802) (Brochu, 1992). Along with this depression, the lateral margin bears an anterolateral straight ridge (anterolateral proximal ridge, according to Leardi et al., 2015), corresponding to the insertion of M. tibialis anterior. The ridge is proximodistally elongated, as in Araripesuchus tsangatsangana, but not Simosuchus clarki, which bears a tuberosity in the corresponding area (Turner, 2006; Sertich & Groenke, 2010). Caudally (Figs. 9A–9B), the lateral and medial articular facets are separated by a small notch, the “fossa flexoria” sensu Hutchinson (2002) or “posterior cleft” sensu Sertich & Groenke (2010). In Pissarrachampsa sera this fossa is more excavated, as in Araripesuchus tsangatsangana and Stratiotosuchus maxhechti, than in Sebecus icaeorhinus, Yacarerani boliviensis, and Alligator mississippiensis (Brochu, 1992; Turner, 2006; Riff & Kellner, 2011; Pol et al., 2012; Leardi et al., 2015).

The tibial shaft is smooth and rounded in cross section, and craniolaterally bowed. This bowing (see character 336 of Leardi, Fiorelli & Gasparini, 2015) can be seen in different degrees within Mesoeucrocodylia. In Pissarrachampsa sera, Baurusuchus albertoi, Stratiotosuchus maxhechti, and Sebecus icaeorhinus the shaft is markedly bowed, differing from the slightly bowed tibia of Yacarerani boliviensis, Simosuchus clarki, and Araripesuchus tsangatsangana, or the straight one in Alligator (Pol et al., 2012; Leardi et al., 2015). There is no distinguished torsion in the tibial shaft of Pissarrachampsa sera. In caudal view (Figs. 9A–9B), it bears a faint ridge for the insertion of M. flexor digitorum longus. This structure is more prominent in other baurusuchids, such as Stratiotosuchus maxhechti and Baurusuchus albertoi, but absent in Araripesuchus tsangatsangana (Turner, 2006; Nascimento & Zaher, 2010; Riff & Kellner, 2011). In extant crocodylians, the longitudinal crest can be marked (e.g., Alligator and Melanosuchus), or slightly prominent (Caiman).

The distal expansion of the tibia is divided inti lateral and medial portions, both contacting the astragalus. The medial portion is distally projected, forming an oblique distal margin relative to the transverse plane. A similar condition is seen in other mesoeucrocodylians, such as Sebecus icaeorhinus, Stratiotosuchus maxhechti, Notosuchus terrestris, Araripesuchus tsangatsangana, and Yacarerani boliviensis (Turner, 2006; Fiorelli & Calvo, 2008; Riff & Kellner, 2011; Pol et al., 2012; Leardi et al., 2015), and it is different from the sub-equally expanded distal tibia of living crocodylians (Alligator and Crocodylus), and also some notosuchians like Simosuchus clarki, Mariliasuchus amarali, and Microsuchus schilleri (Brochu, 1992; Sertich & Groenke, 2010; Nobre & Carvalho, 2013; Leardi, Fiorelli & Gasparini, 2015). In distal view, the tibial surface has a crescentic shape, resembling more the pattern seen in Araripesuchus tsangatsangana and Yacarerani boliviensis, than the “L-shaped” pattern of Sebecus icaeorhinus (Turner, 2006; Pol et al., 2012; Leardi et al., 2015). The craniolateral margin of the distal portion of the tibial expansion is curved, followed by a short and sharp crest that ends caudally at the fibular contact (Fig. 9B, indicated by an arrow). A triangular depression is seen at the caudal surface between the medial and lateral edges of this expansion. First described for Araripesuchus tsangatsangana (Turner, 2006), this structure is well excavated in other mesoeucrocodylians, such as Sebecus icaeorhinus, Stratiotosuchus maxhechti, and Mariliasuchus amarali (Pol et al., 2012; Riff & Kellner, 2011; Nobre & Carvalho, 2013), but relatively shallow in Baurusuchus albertoi and Yacarerani boliviensis (Nascimento & Zaher, 2010; Leardi et al., 2015). Extant crocodylians, such as Caiman, show a clear depression in the same area, but this structure is not triangular. Cranially, close to the medial margin of the distal expansion, there is a protuberance for insertion of M. interosseus cruris. This structure is placed more proximally in extant taxa, slightly developed in Caiman and Melanosuchus, but marked in Alligator (Brochu, 1992). Among Baurusuchidae, both Stratiotosuchus maxhechti and Baurusuchus albertoi bear the same protuberance, although less prominent in the latter (Nascimento & Zaher, 2010; Riff & Kellner, 2011). Craniolaterally, the distal end of the tibia is devoid of the circular depression for the attachment of the medial tibioastragalar ligament, which is clearly seen in Araripesuchus tsangatsangana (Turner, 2006).

Fibula.

Both fibulae of the holotype of Pissarrachampsa sera (LPRP/USP 0019) are virtually complete (Fig. 9) and in articulation with the tibiae. This is also the case for the fibula of LPRP/USP 0744, preserved in articulation with femur and tibia. The fibula of the holotype is 17 cm long, slender and slightly shorter than the tibia. The fibular width corresponds to half that of the tibia, differing from Baurusuchus albertoi, the fibula of which is three times thinner than the tibia (Nascimento & Zaher, 2010). The proximal articular surface is gently concave, with the lateral border more developed than the medial. In proximal view, the fibula is crescentic in shape and the medial margin is slightly notched. In contrast, the proximal fibula of Stratiotosuchus maxhechti is caudally wedged (Riff & Kellner, 2011).

The proximal end of the fibula is lateromedially flat and strongly expanded caudally. The living forms Melanosuchus, Caiman, and Alligator, bear the same caudal expansion for the attachment of the long external lateral ligament (Brochu, 1992), which is also present in baurusuchids such as Stratiotosuchus maxhechti and Baurusuchus albertoi (Nascimento & Zaher, 2010; Riff & Kellner, 2011). Indeed, the shape of the proximal fibular end varies systematically within Crocodyliformes (Turner, 2006). Whereas modern crocodylians, such as Alligator, bear a straight caudal margin, Yacarerani boliviensis, Araripesuchus tsangatsangana, and Araripesuchus gomesii have strongly inflected caudal margin (Turner, 2006; Leardi et al., 2015), baurusuchids have an intermediate condition, with the caudal margin of the proximal head is slightly curved. Proximocranially, there are attachment scars for M. flexor digitorius longus. The lateral iliofibularis trochanter is sharply raised and proximodistally elongated (Figs. 9C–9F), differing from that of Stratiotosuchus maxhechti, Baurusuchus albertoi, Araripesuchus tsangatsangana, and Yacarerani boliviensis, in which the iliofibularis trochanter is shorter and does not reach the proximal edge (Turner, 2006; Nascimento & Zaher, 2010; Riff & Kellner, 2011; Leardi et al., 2015). In extant forms, this trochanter is tubercle-shaped and distant from the proximal edge (Brochu, 1992).

The fibular shaft is almost entirely compressed lateromedially, except in its middle portion, which is elliptical in cross-section. Laterally, the fibular shaft bears faintly developed ridges, as in Baurusuchus albertoi, corresponding to the origin of M. peroneus longus (sensu Brochu, 1992) or M. fibularis longus (sensu Hutchinson, 2002). A different condition is seen in Stratiotosuchus maxhechti, in which that ridge is well developed (Riff, 2007). Among extant crocodylians, both Caiman and Melanosuchus show weakly developed ridges on the lateral surface of the fibular shaft, whereas in Alligator the fibula bears well developed crests and a slightly rugose shaft lateral surface (Brochu, 1992). In medial view, the shaft is mostly smooth and lacks any distinctive muscle scar. However, the caudodistal surface is rugose, revealing scars possibly related to the attachment for M. interosseus cruris, as also observed in Araripesuchus tsangatsangana and Stratiotosuchus maxhechti (Turner, 2006; Riff, 2007). There is a small vascular foramen on the caudal surface near the midshaft. The tibial distal end is enlarged with a triangular distal outline, as in Araripesuchus tsangatsangana and Microsuchus schilleri (see Leardi, Fiorelli & Gasparini, 2015: character 425). As in Alligator, Caiman, and Melanosu“hus, a “dis”al hook” (sensu Brochu, 1992) contacts the tibia and tapers medially. This differs from the condition in Stratiotosuchus maxhechtiand Yacarerani boliviensis, in which the medial end of the distal margin of the tibia is rounded (Riff & Kellner, 2011; Leardi et al., 2015). The contact of the distal hook with the tibia is more proximal then the distal tibial articulation (Fig. 9), and differs from the pattern in Microsuchus schilleri, the distal hook of which contacts the tibia more distally. This hook is absent in Araripesuchus tsangatsangana and Yacarerani boliviensis (Turner, 2006; Leardi et al., 2015).

Figure 10 Pissarrachampsa sera (holotype, LPRP/USP 0019), photographs and schematic drawings of the left astragalus and calcaneum in proximal (A and B), cranial (C and D), and distal views (E and F).

Abbreviations: aho, “anterior hollow”; cbc, cranial body of calcaneum; ctc, caudal tuber of calcaneum; fif, fibular facet; lch, lateral channel; lrc, lateral ridge of calcaneal tuber; ltb, lateral tubercule; ltf, lateral tibial facet; m i, ii?, area for articulation with metatarsals I and II; mch, medial channel; mdr, medial distal roller; mfl, medial flange; mrc, medial ridge of calcaneal tuber; mtf, medial tibial facet; pat, pit for astragalar -tarsal ligament; peg, astragalar peg; td iv?, area for the articulation with tarsal distal IV. Scale bar equals 2 cm.

Tarsus.

Both complete astragali and calcanea are preserved in articulation (Fig. 10) in the holotype of Pissarrachampsa sera (LPRP/USP 0019), although the more distal tarsal bones are not preserved. The best preserved left astragalus and calcaneum are slightly displaced from their original positions. The tarsal morphology of Pissarrachampsa sera is similar to that of other crocodylomorphs with the “crocodile normal” condition, in which the astragalar “peg” fits into the calcaneal “socket” (Chatterjee, 1978; Chatterjee, 1982). In this configuration, the astragalus is fixed in articulation with tibia and the ankle rotation occurs between astragalus and calcaneum (Brochu, 1992).

Proximally, the astragalus bears a concave and laterally elongate surface for articulation with the distal tibia (Figs. 10A–10B). The division of this surface for the reception of medial and lateral condyles of the tibia is weak and both facets are similar in lateromedial extension. These are bounded caudally by a ridge, but this structure is more developed on the lateral region of the medial tibial facet. As in the baurusuchids Baurusuchus albertoi and Stratiotosuchus maxhechti and the sebecid Sebecus icaeorhinus (Riff & Kellner, 2011; Pol et al., 2012), there is no sign of an “astragalar fossa” (Hecht & Tarsitano, 1984). This differs from the morphology of extant taxa, Simosuchus clarki, and Yacarerani boliviensis, in which the fossa is present and well developed (Hecht & Tarsitano, 1984; Brochu, 1992; Sertich & Groenke, 2010; Leardi et al., 2015). The lateral tibial facet is flat, equally developed lateromedially and ends just craniomedial to the fibular facet (Figs. 10A–10D). The lateromedial edge of the lateral tibial facet seems to lack the notch observed in Yacarerani boliviensis, Stratiotosuchus maxhechti, Sebecus icaeorhinus, and Lomasuchus palpebrosus (Gasparini, Chiappe & Fernandez, 1991), but this surface is damaged in both left and right elements (Pol et al., 2012; Leardi et al., 2015). The lateral tibial and fibular articular surfaces are set almost perpendicular to each other, as in other fossil crocodyliforms, such as Simosuchus clarki, Baurusuchus albertoi, Stratiotosuchus maxhechti, Yacarerani boliviensis, and also in extant forms (Hecht & Tarsitano, 1984; Brochu, 1992; Nascimento & Zaher, 2010; Sertich & Groenke, 2010; Riff & Kellner, 2011; Leardi et al., 2015). The medial tibial articular facet is reniform, as in Sebecus icaeorhinus, but more craniocaudally expanded, as in Simosuchus clarki and Yacarerani boliviensis (Sertich & Groenke, 2010; Leardi et al., 2015). The fibular facet is trapezoidal and slightly concave. Distally, the astragalus bears a medial distal roller (Hecht & Tarsitano, 1984) and the calcaneal articulation (Brochu, 1992). The distal roller is elliptical in distal view and extends cranioproximally merging into the craniomedial edge of the tibial facet. The metatarsals are not preserved in articulation with the astragali, but there is a slight depression in the distal surface of the medial distal roller that is probably related to the articulation of both first and second metatarsals, as in Baurusuchus albertoi, Simosuchus clarki, Stratiotosuchus maxhechti, and extant forms (Hecht & Tarsitano, 1984; Nascimento & Zaher, 2010; Sertich & Groenke, 2010; Riff & Kellner, 2011).

The calcaneal articulation is formed by a well developed distolaterally directed peg as in other crocodyliforms. This is divided in two distinct areas, the distal area of articulation (“astragalar trochlea” of Hecht & Tarsitano, 1984) and the lateral articular surface. Yet, the morphology of these facets cannot be accessed due the tight articulation with the calcaneum in both sides. The cranial surface of the astragalus consists of a limited non-articular region (the “anterior hollow” of Hecht & Tarsitano, 1984). This area is more restricted when compared to that of Sebecus icaeorhinus, Simosuchus clarki, and extant forms, but similar to the condition of Baurusuchus albertoi and Stratiotosuchus maxhechti (Hecht & Tarsitano, 1984; Brochu, 1992; Nascimento & Zaher, 2010; Sertich & Groenke, 2010; Riff & Kellner, 2011; Pol et al., 2012). As in Sebecus icaeorhinus, Stratiotosuchus maxhechti, and Simosuchus clarki (Pol et al., 2012; Leardi et al., 2015), the “anterior hollow” does not seem bounded distally and laterally by crests, but its lateralmost surface is somewhat damaged. Distally, the pit for the astragalar-tarsale ligament is located at the anterior hollow, close to the medial distal roller (Brinkman, 1980). The pit is well-developed, as in Yacarerani boliviensis, Simosuchus clarki, Stratiotosuchus maxhechti, and Sebecus icaeorhinus, differing from the reduced depression of Baurusuchus albertoi (Sertich & Groenke, 2010; Nascimento & Zaher, 2010; Riff & Kellner, 2011; Pol et al., 2012; Leardi et al., 2015). The vascular foramina observed in other taxa, such as Baurusuchus albertoi, Stratiotosuchus maxhechti, and Simosuchus clarki (Nascimento & Zaher, 2010; Sertich & Groenke, 2010; Riff & Kellner, 2011), are not present in Pissarrachampsa sera, nor in Sebecus icaeorhinus (Pol et al., 2012).

The calcaneum of Pissarrachampsa sera is robust and mediolaterally developed, as in Yacarerani boliviensis, Baurusuchus albertoi, Stratiotosuchus maxhechti, and Sebecus icaeorhinus, differs from the mediolaterally compressed calcaneum of Araripesuchus tsangatsangana and Uruguaysuchus aznarezi (Rusconi, 1933) (Turner, 2006; Nascimento & Zaher, 2010; Sertich & Groenke, 2010; Riff & Kellner, 2011; Pol et al., 2012; Leardi et al., 2015). It is formed by a cranial body, a socket for the reception of the astragalar peg, and the caudally directed tuber (Brochu, 1992). As in other crocodyliforms, the cranial body in P. sera contacts the astragalus, fibula, and possibly the fourth distal tarsal (Brinkman, 1980; Hecht & Tarsitano, 1984; Brochu, 1992; Sertich & Groenke, 2010; Pol et al., 2012).

The cranial and proximal portions of the cranial body form a well-developed rounded articular surface (a roller) that articulates medially with the astragalus and proximally with the fibula. This morphology is widespread, also seen in living forms and other fossil crocodylians, as Baurusuchus albertoi, Stratiotosuchus maxhechti, Sebecus icaeorhinus, Simosuchus clarki, and Araripesuchus tsangatsangana (Brinkman, 1980; Turner, 2006; Sertich & Groenke, 2010; Nascimento & Zaher, 2010; Riff & Kellner, 2011; Pol et al., 2012). No ridge is present at the articular surface of the roller, which in Simosuchus clarki separates the medial articulation area for the astragalus and the lateral articulation area for the fibula (Sertich & Groenke, 2010). This rounded surface slopes abruptly cranioventrally, forming a distally directed surface, which probably contacted the fourth distal tarsal. In Pissarrachampsa sera, this surface is flat and elliptical in distal view, resembling the condition in Stratiotosuchus maxhechti (Riff & Kellner, 2011). The lateral portion of the cranial body forms a well-developed flat surface that lacks any articular facet. This surface is proximodistally restricted and does not overcome the proximodistal extension of the distal tuber. The medial face of the cranial body forms the calcaneal socket. Most of the morphology of this area is not accessible due the articulation with the astragalus, but a faint medial flange overhangs the calcaneal socket as in Simosuchus clarki (Sertich & Groenke, 2010).

The calcaneal tuber is caudally directed and sub-elliptical in caudal view, as in Baurusuchus albertoi and Stratiotosuchus maxhechti (Nascimento & Zaher, 2010; Riff & Kellner, 2011). The caudal surface of the tuber is orthogonal to the distal facet of the calcaneal condyle, and is deeply concave, forming a slot for attachment of M. gastrocnemius (Brochu, 1992; Leardi et al., 2015). The concavity divides the tuber into well-marked lateral and medial ridges, as in Baurusuchus albertoi, Stratiotosuchus maxhechti, Sebecus icaeorhinus, Araripesuchus tsangatsangana, and Simosuchus clarki (Turner, 2006; Riff & Kellner, 2011; Sertich & Groenke, 2010; Pol et al., 2012). Unlike in Stratiotosuchus maxhechti, there is no transversal ridge separating the caudal surface in proximal and distal areas (Riff & Kellner, 2011). The lateral ridge is shorter than the medial one, as in Simosuchus clarki and Uruguaysuchus aznarezi, whereas in other taxa (Baurusuchus albertoi, Stratiotosuchus maxhechti, Sebecus icaeorhinus) both ridges are equally developed (Sertich & Groenke, 2010; Nascimento & Zaher, 2010; Riff & Kellner, 2011; Pol et al., 2012). The lateral ridge bears a lateral tubercle, as in Yacarerani boliviensis, Sebecus icaeorhinus and Stratiotosuchus maxhechti (Riff & Kellner, 2011; Pol et al., 2012; Leardi et al., 2015). The tubercle extends laterodistally and invades the lateral surface of the calcaneal tuber (Figs. 10E–10F). A well-defined groove flanks the medial side of the calcaneal tuber. This corresponds to the “medial channel” of Hecht & Tarsitano (1984). It expands proximolaterally in a shallow and wide surface that terminates abruptly at the lateral edge of the calcaneum. A lateral groove also separates the distal articular surface of the cranial body from the calcaneum tuber, just medial to the lateral tubercle, as seen in Simosuchus clarki (Sertich & Groenke, 2010).

Pes.

Pissarrachampsa sera has three preserved pedes, the left pes of the holotype (LPRP/USP 0019) and two referred (a left and a right) pedes (LPRP/USP 0739 and LPRP/USP 0746). The holotype pes is represented by four articulated metatarsals (Fig. 11B), whereas LPRP/USP 0739 includes four isolated metatarsals, and LPRP/USP 0746 comprises four partially preserved articulated digits (Fig. 11A). Metatarsal V is not preserved in any of the specimens of Pissarrachampsa sera, following the trend of reduction of that metatarsal towards Crocodylomorpha (Parrish, 1987). Therefore, the four metatarsals preserved in Pissarrachampsa sera constitute the entire number of fully functional pedal digits, as in all living crocodylians and most fossil crocodyliforms (Riff, 2007).

The metatarsals of Pissarrachampsa sera are longer than the metacarpals, as in Baurusuchus albertoi, Araripesuchus tsangatsangana, Stratiotosuchus maxhetchi, Simosuchus clarki and Yacarerani boliviensis (Turner, 2006; Nascimento & Zaher, 2010; Sertich & Groenke, 2010; Riff & Kellner, 2011; Leardi et al., 2015). Moreover, metatarsals II and III are slightly longer than metatarsals I and IV, as in Baurusuchus albertoi and possibly in Yacarerani boliviensis and S. maxhetchi (Nascimento & Zaher, 2010; Riff & Kellner, 2011; Leardi et al., 2015). The proximal articular surfaces of the metatarsals are lateromedially expanded, especially in their lateral margin. As a result, the proximal surface of each metatarsal overlaps the medial portion of the proximal surface of the immediate lateral metatarsal (Fig. 11—LPRP/USP 0746) as in Baurusuchus albertoi, Simosuchus clarki, and Stratiotosuchus maxhetchi (Nascimento & Zaher, 2010; Sertich & Groenke, 2010; Riff & Kellner, 2011). This morphology is different from that of Araripesuchus tsangatsangana, in which a medial expansion of these surfaces underlies the proximal surface of the immediate medial metatarsal, and from Yacarerani boliviensis, in which there is a medial expansion of the surface in each metatarsal that overlaps the immediate medial metatarsal (Turner, 2006; Leardi et al., 2015). The distal articular surfaces are divided by a groove in the medial and lateral condyles, as in Simosuchus clarki, Baurusuchus albertoi and Stratiotosuchus maxhechti (Nascimento & Zaher, 2010; Sertich & Groenke, 2010; Riff & Kellner, 2011).

Figure 11 Pissarrachampsa sera, photographs of two pedes and ungual phalanges. (A) right pes of LPRP/USP 0746 in ventral view; (B) left pes of LPRP/USP 0019 (holotype) in dorsal view. (C) ungual phalanges of LPRP/USP 0019 (holotype).

Abbreviations: I mt, metatarsal I; II mt, metatarsal II; III mt, metatarsal III; IV mt, metatarsal IV; ast, astragalus; dph, distal phalanx; mph, medial phalanx; pph, proximal phalanx; uph, ungueal phalanx. Scale bar equals 5 cm.

Only LPRP/USP 0746 preserves articulated phalanges (Fig. 11A), but the phalangeal formula cannot be assessed. The phalanges have a blocky appearance and a constriction between the expanded proximal and distal ends, as in Simosuchus clarki, Baurusuchus albertoi, Stratiotosuchus maxhechti, and Araripesuchus tsangatsangana (Turner, 2006; Nascimento & Zaher, 2010; Sertich & Groenke, 2010; Riff & Kellner, 2011). The proximal phalanges preserved in LPRP/USP 0746 are relatively longer than those preserved in the right manus of the holotype (both hands are similar in size), a pattern described for both Baurusuchus albertoi and Stratiotosuchus maxhechti (Nascimento & Zaher, 2010; Riff & Kellner, 2011). Also, the proximal phalanges preserved in LPRP/USP 0746 are longer than the preserved more distal phalanges, as in Baurusuchus albertoi, Araripesuchus tsangatsangana, and S. maxhetchi (Turner, 2006; Nascimento & Zaher, 2010; Riff & Kellner, 2011).

Aside from the articulated phalanges of LPRP/USP 0746, three disarticulated pedal ungual phalanges were found associated with the holotype skeleton (Fig. 11C). They decrease in size from the first to the third digit, as in Baurusuchus albertoi, Stratiotosuchus maxhechti, Uberabasuchus terrificus and living crocodylians (Müller & Alberch, 1990; Vasconcellos, 2006; Riff, 2007; Nascimento & Zaher, 2010). They form curved claws, with a robust base, and bear foramina in both lateral and medial surfaces, as also present in Baurusuchus albertoi and, possibly, in Araripesuchus tsangatsangana (Turner, 2006; Nascimento, 2008; Nascimento & Zaher, 2010).

Results and Discussion

Body size and mass estimates of Pissarrachampsa sera

The preserved elements of the holotype (LPRP/USP 0019), particularly the femora, allow estimating the body size and mass of Pissarrachampsa sera. Based on the protocol presented by Farlow et al. (2005), we estimated that Pissarrachampsa sera had a total length varying between 2.7 and 3.5 m, and a body mass between 81 and 163 kilograms (for detailed results see Supplemental Information). This significant variation is also observed in estimates for other terrestrial crocodyliforms, such as Protosuchus and Sebecus (Farlow et al., 2005; Pol et al., 2012). The regressions of Farlow et al. (2005) were built with data from Alligator mississippiensis, and might not be as accurate as desired for fossil taxa with different habits and body proportions, as already pointed out by other works (e.g., Young et al., 2011; Pol et al., 2012).

Indeed, the comparison with nearly complete baurusuchid specimens permits assessing the accuracy of these regressions for the group. Comparisons to more complete baurusuchids such as the 1.9 m long specimen referred to Baurusuchus salgadoensis (lacking only the skull and pectoral girdle), the 1.3 m long holotype of Baurusuchus albertoi (lacking the tip of tail and snout), and the 1.1 m long holotype of Aplestosuchus sordidus (lacking the tail) (Nascimento, 2008; Vasconcellos & Carvalho, 2010; Godoy et al., 2014) suggest that it is unlikely that any of these specimens reached the maximum length estimated for Pissarrachampsa sera (3.49 m) using the regressions. Further, after applying the formulas from Farlow et al. (2005) for Baurusuchus albertoi and B. salgadoensis (both with femora well preserved), we obtained a total length of approximately 3.8 m for both taxa (see Supplemental Information). Even though not completely preserved, this is evidence that, at least for baurusuchids, these regressions are overestimating the size of the specimens. Additionally, in order to test the validity of the mass estimates obtained with the formulas from Farlow et al. (2005), we also applied the equations presented by Campione & Evans (2012), which uses proximal (stylopodial) limb bone circumference to obtain total body mass, and seems to work well for many fossil taxa (e.g., Castanhinha et al., 2013; Benson et al., 2014; Reisz & Fröbisch, 2014). After applying the femur-based equation, the mass estimate obtained for Pissarrachampsa sera was approximately 71 kilograms, lower than the lowest value obtained using Farlow et al. (2005) formulas.

Regardless of the incompleteness of specimens and inaccuracy of size estimates, it is very likely that an adult individual of Pissarrachampsa sera reached at least 2 m (Fig. 12), placing the taxon amongst the largest terrestrial predators of Late Cretaceous environments in southeast Brazil, together with other baurusuchids and theropods (Riff & Kellner, 2011; Godoy et al., 2014). The Bauru Group rocks have provided numerous carnivorous crocodyliforms (e.g., Campos et al., 2001; Carvalho, Campos & Nobre, 2005; Godoy et al., 2014), particularly baurusuchids, and many titanosaur sauropods (e.g., Kellner & Azevedo, 1999; Salgado & Carvalho, 2008; Santucci & Arruda-Campos, 2011), but very few theropods (Novas et al., 2008; Bittencourt & Langer, 2011; Méndez, Novas & Iori, 2012; Azevedo et al., 2013). This has been used as evidence for the rearrangement of roles in this paleoecosystem, with baurusuchids occupying the typical ecological niche of theropods or at least competing for the same niche (Gasparini, Fernandez & Powell, 1993; Candeiro & Martinelli, 2006; Riff & Kellner, 2011). However, although the morphology of baurusuchids indicates a highly specialized predatory habit, similar to that of theropods, it seems unlikely that even larger baurusuchids could have preyed on adult sauropods (>8-meter length for some titanosaurs; Salgado & Carvalho, 2008), if assumed as solitary predators. Although young theropods could have had similar diets to baurusuchids, the morphological differences are also indicative of distinct feeding (Martinelli et al., 2013). Indeed, this hypothesis is supported by the single reliable and identifiable direct evidence of predation among baurusuchids, in which a small sphagesaurid (Mesoeucrocodylia, Notosuchia) was found in the abdominal cavity of the holotypic skeleton of Aplestosuchus sordidus (Godoy et al., 2014). As such, if adult sauropods had any predator in this Cretaceous ecosystem, theropods remain as the most likely ones, and the scarcity of theropods might reflect incomplete or biased sampling. Accordingly, some niche partitioning may have occurred, with baurusuchids preying on smaller animals, as well as young or hatchling sauropods, and adult theropods being able to prey on larger individuals.

Figure 12 Skeletal reconstruction of Pissarrachampsa sera, including all known cranial and postcranial material.

Scale bar equals 80 cm.

Terrestriality in Pissarrachampsa sera

A series of anatomical features have been recognized as related to the terrestrial habits of Crocodyliformes, many of which are observed in the postcranial skeleton of Pissarrachampsa sera. Most of these concern an upright posture and gait, with the limbs held under the body rather than to the side as in extant crocodylians. A characteristic presumably linked to terrestriality is the reduced space between articulated ulna and radius in Pissarrachampsa sera. Although contrasting with the relatively large space in extant crocodylians, this pattern is also observed in other baurusuchids, such as Stratiotosuchus maxhechti and Baurusuchus albertoi, as well as in the terrestrial notosuchian Araripesuchus tsangatsangana (Brochu, 1992; Turner, 2006; Nascimento & Zaher, 2010; Riff & Kellner, 2011). Similarly, the space between tibia and fibula of Pissarrachampsa sera is also reduced. Further, the proximal portion of its tibia bears a well-protruded medial facet that corresponds to the articulation with the tibial condyle of the femur. The uneven proximal facets rotate the distal tibia laterally when in articulation with the femur. Accordingly, both propodium and epipodium were arranged on the same long axis (in caudal or cranial views), allowing a parasagittal movement of the leg during locomotion. This condition is also seen in the terrestrial notosuchians Sebecus icaeorhinus and Simosuchus clarki (Sertich & Groenke, 2010; Pol et al., 2012). The proximal articulation facets of the tibia are caudally separated by an excavated fossa flexoria, and cranially, by a large tuberosity for the insertion of M. flexor tibialis internus.This is evidence of a tight/stable knee joint in agreement with an erect posture. Also, the distal tibial articulation of Pissarrachampsa sera is obliquely disposed, with a more enlarged medial facet, as in Stratiotosuchus (Riff & Kellner, 2011). Extant crocodylians, on the other hand, bear equally developed distal ends (medial and lateral) of the tibia, allowing a range of sprawling to semi-erect high walk (Brinkman, 1980; Parrish, 1986; Parrish, 1987; Gatesy, 1991). This oblique articulation and the sharp distal end of the tibia fits tightly with the astragalus, and can reduce the range of movements. But it also indicates a stable articulation with the foot, allowing some lateral displacement, matching the medial displacement of the distal tibia, denoting an upright posture. This is similar to the ankle articulation morphology seen in the terrestrial sphenosuchians and protosuchians (Parrish, 1987), but it is also observed in more closely-related taxa, as Araripesuchus tsangatsangana and Sebecus icaeorhinus.

Additionally, the less curved femur of P. sera, in comparison to that of living crocodylians, is in accordance with a more erect posture. The faint curvature in this bone is similiar to that seen in Stratiotosuchus maxhechti, for which a parasagittal posture was also claimed (Riff & Kellner, 2011). Hutchinson (2001b) argues that limb bones, such as the femur, with a less accentuate curvature are subjected to bending stresses rather than torsional stresses. That anatomical acquisition would then be related to a more erect posture and terrestrial habits in the archosaurian lineage, whereas bones under torsional stresses, such as sigmoid femora, are associated with forms with a sprawling posture. Still, some of features pointed out by Parrish (1987) as linked to a parasagittal posture in archosaurians are also observed in Pissarrachampsa sera, such as a well-developed and medially inturned femoral head, prominent caudally oriented femoral condyles, and a conspicuous fibular condyle (or lateral condyle). Further, the femur orientation is compatible with the morphology of the ilium of P. sera. The laterally projected and enlarged supraacetabular crest would make it impossible for the femur to be strictly laterally oriented (Riff & Kellner, 2011), but would be compatible with a vertical orientation of a parasagital posture. Still in the pelvic girdle, Pissarrachampsa sera possess a tubercle on the lateral surface of the ischium, located in the attachment area of M. pubioischiotibialis. Riff & Kellner (2011) pointed out that this tubercle is absent in extant forms, and its big size in Stratiotosuchus, similar to the morphology observed in P. sera, can indicate that this muscle was more developed in the baurusuchids. Indeed, Reilly & Blob (2003) show that, in Alligator, this muscle is activated during the “high-walk” locomotion mode, which is compatible with the interpretation of Riff & Kellner (2011) suggesting that a greater development of the M. pubioischiotibialis is compatible with a permanent parasagital posture, more related to a terrestrial lifestyle.

The lack of osteoderms in Pissarrachampsa sera

Pissarrachampsa sera is represented by a series of specimens all from the same locality. The specimens range from the relatively complete and fairly articulated holotype to isolated fragmentary cranial and postcranial elements. So far, no osteoderm was found associated with these specimens, neither elsewhere in the type locality. This raises the question whether the lack of osteoderms represents a taphonomic signature or a genuine anatomical feature of the taxon. In the latter case, Pissarrachampsa sera would be the first terrestrial crocodyliform to completely lack any body armor, with biomechanical implications to be explored.

The specimens of Pissarrachampsa sera were collected without rigorous taphonomic control, but there is geological and paleontological evidence that supports the absence of osteoderms as unrelated to taphonomy. The type locality of P. sera is assigned to the Adamantina Formation and the deposition of this geological unity is associated with arid to semi-arid conditions (Fernandes & Coimbra, 1996; Fernandes & Coimbra, 2000; Batezelli, 2015). In the same way, the local geology suggests a developed paleosol profile that is also indicative of arid to semi-arid conditions (JCA Marsola et al., unpublished data). In this scenario, the prolonged periods without sedimentation lead to erosion and pedogenesis. Furthermore, well-preserved and complete crocodyliform egg clutches are found in the same levels of the body fossils of Pissarrachampsa sera (Marsola, Montefeltro & Langer, 2011). Crocodyliform eggs are particularly fragile to long-range transport (Grellet-Tinner et al., 2006; Hayward et al., 2000), whereas the skeletal elements of P. sera do not show significant signs of abrasion caused by transport (Montefeltro, Larsson & Langer, 2011). Therefore, the decay and burial of the P. sera remains most likely occurred in a low-energy, probably sub-aerial environment.

Araújo-Júnior & Marinho (2013) analyzed the taphonomy of one specimen of Baurusuchus pachecoi from the same formation, collected in Jales (São Paulo, Brazil), which matches the putative pre-burial conditions experienced by Pissarrachampsa sera. In that study, osteoderms were found close to their in vivo position, even after being exposed to some degree of scavenging and sub-aerial decay. A similar pattern of osteoderm disarticulation was found by Beardmore et al. (2012) for the marine crocodyliform Steneosaurus (Geoffroy Saint-Hilaire, 1825), from the Posidonienschiefer Formation (Lower Jurassic, Germany), which decayed and were buried in a quiet-water, marine basin. In that case, osteoderms are placed close to the carcass even in specimens with greater degree of disarticulation. The same pattern is seen in actualistic taphonomic experiments in juvenile Crocodylus porosus (Schneider, 1801), in which the osteoderms remain at the vicinity of the carcass even with relatively prolonged subaerial and subaqueous decay (Syme & Salisbury, 2014, Fig. 6). In fact, a series of fossil crocodyliforms, both close and distantly related to Pissarrachampsa sera, are recovered with associated osteoderms, even showing a relatively advanced degree of disarticulation, as. Susisuchus anatoceps (Salisbury et al., 2003), Candidodon itapecuruense (Carvalho & Campos, 1988) (Nobre, 2004), Simosuchus clarki (Krause et al., 2010), Alligatorellus (Gervais, 1871) (Schwarz-Wings et al., 2011), Wannchampsus kirpachi (Adams, 2014), Diplocynodon (Pomel, 1847) (Hastings & Hellmund, 2015), and Caipirasuchus montealtensis (Andrade & Bertini, 2008) (Iori, Carvalho & Marinho, 2016). We took into consideration the possibility that Pissarrachampsa sera had its osteoderms disarticulated earlier in the decay process. This is possible and is supported by specimens of closely-related notosuchians with fairly articulated postcrania but lacking osteoderms, such as Mariliasuchus amarali (UFRJ-DG-50-R), Notosuchus terrestris (MUCPv-137), Sebecus icaeorhinus (Pol et al., 2012). However, in the particular case of P. sera we regard this as unlikely, given the complete absence of these elements in the entire outcrop and the number of specimens recovered. Therefore, in light of all evidence we suggest the lack of osteoderms is an inherent and diagnostic feature of Pissarrachampsa sera.

The presence of osteoderms is considered plesiomorphic for Crocodyliformes (Scheyer & Desojo, 2011), as these structures are found in most pseudosuchians (Brown, 1933; Wu & Chatterjee, 1993; Clark & Sues, 2002; Sues et al., 2003; Pol & Norell, 2004; Clark, 2011; Nesbitt, 2011; Scheyer & Desojo, 2011). Likewise, this ancestral condition is inferred for most internal nodes of Crocodyliformes, which bear at least one pair of parasagittal rows forming the body armor (Salisbury & Frey, 2001; Frey & Salisbury, 2001; Hill, 2005; Pierce & Benton, 2006; Jouve et al., 2006; Marinho & Carvalho, 2009; Pol, Turner & Norell, 2009; Hill, 2010; Andrade et al., 2011; Pol et al., 2012; Nobre & Carvalho, 2013; Tennant & Mannion, 2014). The only exception known so far is the complete absence of osteoderms in the marine metriorhynchids, a feature probably associated with their aquatic lifestyle (Young et al., 2010; Young et al., 2013; Molnar et al., 2015). Similarly, metriorhynchids do not have palpebral bones roofing the orbits (Nesbitt, Turner & Weinbaum, 2012), and previous analyses of the crocodylian skeletogenesis show that postcranial osteoderms match the palpebral development (Vickaryous & Hall, 2008). In this case, it might have been a common cause underlying the successive loss of the palpebrals and postcranial osteoderms in Thalattosuchia and Metriorhynchidae.

Molnar et al. (2015) presented evidence that the loss of osteoderms in Metriorhynchidae is related to an increasing aquatic adaptation in this group, whereas the rigid series of osteoderms of early crocodylomorphs would be related to terrestrial habits. In this scenario, the presence of non-imbricate osteoderms in teleosaurid thalattosuchians and the more flexible arrangement of these structures in the extant semi-aquatic forms would represent intermediate stages (Salisbury & Frey, 2001; Molnar et al., 2015). The presence of one pair of parasagittal rows of oval osteoderms is considered a plesiomorphic state for Baurusuchidae, as all specimens previously described with postcranial remains exhibit this pattern (Nascimento & Zaher, 2010; Vasconcellos & Carvalho, 2010; Araújo-Júnior & Marinho, 2013; Godoy et al., 2014). The osteoderms of these forms (e.g., Aplestosuchus sordidus) barely imbricate and are not sutured to their counterparts, which might represent an intermediate condition towards the total lack of osteoderms seen in P. sera. The phylogenetic position of P. sera among Pissarrachampsinae, as well as its smaller size when compared to Baurusuchinae, lead to two possible underlying factors for the absence of body armor in this taxon. It could be assigned as a synapomorphy of Pissarrachampsinae and interpreted as a historical factor, also implying the absence in other members of the clade, for which we still do not have information (Campinasuchus dinizi and Wargosuchus australis). Alternatively, if the absence of osteoderms is confirmed in the other smaller and early-diverging taxa, Cynodontosuchus rothi (Woodward, 1896) and Gondwanasuchus scabrosus (Marinho et al., 2013), this condition could be linked to the reduced size of the taxa.

Yet, in both scenarios, the complete absence of osteoderms in P. sera and the reduction of the body armor in other baurusuchids had biomechanical implications, with the osteoderms in other baurusuchids possibly playing a diminutive role in the bracing system and in the sustained terrestrial locomotion of these animals. This is different from what is inferred for other terrestrial Crocodylomorpha such as “sphenosuchians” and the peirosaurids, in which the osteoderms played an important role in the bracing system and sustained erect locomotion (Salisbury & Frey, 2001; Molnar et al., 2015; Tavares, Ricardi-Branco & Carvalho, 2015). One exception to the general pattern is the absence of osteoderms in the “sphenosuchian” Junggarsuchus sloani (Clark et al., 2004). This assertion is supported by the reduced transverse process and the verticalized zygapophyses which imply a bracing system not compatible to the extant forms (Salisbury & Frey, 2001). The preserved vertebrae in P. sera belong to caudal-dorsal postion therefore not overlapping the more cranial vertebrae preserved in Junggarsuchus sloani (Clark et al., 2004). However, the vertebrae of P. sera also have more verticalized zygapophyses suggesting reduced undulating lateral movements in both taxa. On the other hand, the transverse process preserved in P. sera is expanded and more similar to the extant forms than to Junggarsuchus sloani (Salisbury & Frey, 2001; Clark et al., 2004; Molnar et al., 2015). An expanded transverse process is also present in caudal-dorsal vertebrae of metriorhynchids (Young et al., 2013; Molnar et al., 2015). Accordingly, there is no perfect correlation between the occurrence of expanded transverse process and presence of osteoderms in crocodyliforms. In light of the evidence, we suggest that Baurusuchidae in general, and P. sera in particular, acquired a unique bracing system with little or no participation of the osteoderms in the sustained erect locomotion.

Phylogenetic analysis and the significance of postcranial characters in Crocodyliformes phylogeny

Here, for the first time, the postcranial data for Pissarrachampsa sera was included in a phylogenetic analysis. This resulted in scoring a total of 34 additional characters (see the Supplemental Information) for the taxon in the data matrix presented by Leardi, Fiorelli & Gasparini (2015), which is the most recent work including a substantial amount of postcranial characters. The resulting data matrix (439 characters and 111 taxa) was analysed in TNT (Goloboff, Farris & Nixon, 2008a; Goloboff, Farris & Nixon, 2008b) via heuristic searches under the following parameters: 10,000 replicates of Wagner Trees, hold 10, TBR (tree bi-section and reconnection) for branch swapping, and collapse of zero length branches according to “rule 1” of TNT (min.length = 0). The result of our analysis (Supplemental Information) was exactly that presented by Leardi, Fiorelli & Gasparini (2015), and all the clades are supported by the same set of synapomorphies as in the original study.

We also conducted exploratory analyses to investigate the significance of the postcranial anatomy for the phylogenetic relationships of crocodyliforms based on the data matrix used in this study. We created two subsets of the original matrix, one using only cranial characters (315 characters), and another solely with postcranial characters (124 characters). As some of the taxa in this dataset do not have either cranial or post-cranial data, we performed an extra “control analysis” including only taxa for which elements of both subsets of the skeleton are scored. This “control analysis” was performed to test whether simply removing taxa caused an impact on the overall relationships between taxa. A total of 39 taxa (all from the ingroup) were excluded following this criteria (Supplemental Information), and the 72 remaining taxa were used in the two exploratory analyses.

The topology of the strict consensus of the MPT’s obtained in the “control analysis” (Fig. 13) is consistent with that of the original dataset. A single difference in the branching pattern is that the “protosuchians” are less resolved than in the original dataset, but a fully compatible structure is recovered for Mesoeucrocodylia. In the basal dichotomy of this clade, one of the branches leads to Notosuchia, including Uruguaysuchidae, Peirosauridae, and Ziphosuchia, with the latter containing Baurusuchidae and Sebecidae. The other branch leads to Neosuchia, including a clade containing the longirostrine forms (Tethysuchia + Thalattosuchia) and another clade including Atoposauridae, Goniopholididae and Eusuchia. Thus, this result indicates that the deletion of the 39 taxa did not have a significant impact on the inferred relationships.

Figure 13 Strict consensus tree of the “control analysis” after excluding taxa with no cranial or postcranial characters.

Silhouettes of representative crocodylomorphs from Bronzati, Montefeltro & Langer (2012) and Bronzati, Montefeltro & Langer (2015).

The strict consensus tree of the analysis using only cranial characters does not show a great number of polytomies and is similar to the original complete analysis (Leardi, Fiorelli & Gasparini, 2015), even the arrangement of “protosuchians” (Fig. 14), but there are important discrepancies. One is related to the paraphyletic arrangement of taxa retrieved as members of the Notosuchia clade in the original and control analyses. Some clades within Notosuchia (sensu Pol et al., 2012; Pol et al., 2014; Leardi, Fiorelli & Gasparini, 2015; Leardi et al., 2015), such as Sphagesauridae, Uruguaysuchidae, and Baurusuchidae, are still grouping in a more inclusive clade, but sebecids and peirosaurids are more closely related to neosuchians than to other notosuchians. Still, a monophyletic Sebecia (Peirosauridae + Sebecidae) is recovered in this exploratory analysis, recovering a pattern proposed by previous works (Larsson & Sues, 2007; Montefeltro et al., 2013). The clade Sebecia was supported by anatomical similarities of the palate of both peirosaurids and sebecids, which in the absence of postcranial characters, favour the recovery of this relationship.

Figure 14 Strict consensus tree of the analysis based only on cranial characters.

Name of clades between quotes indicates that their inclusivity differs from those of the “control analysis.” Clade with the node marked by a square (Sebecia) represents those not present in the “control analysis.” Silhouettes of representative crocodylomorphs from Bronzati, Montefeltro & Langer (2012) and Bronzati, Montefeltro & Langer (2015).

Additional differences are in the internal relationships of Neosuchia. Despite the presence of monophyletic Goniopholididae, Tethysuchia, Thalattosuchia, and Atoposauridae, substantial changes are noted, as Eusuchia is paraphyletically arranged in relation to Tethysuchia + Thalattosuchia. The recovery of the clade encompassing Tethysuchia and Thalattosuchia probably reflects the major modifications on the skull of longirostrine forms belonging to these groups.

The results were very different when the analysis was conducted only with postcranial characters. The strict consensus is poorly resolved (Supplemental Information). A strict consensus tree with low resolution can occur for distinct reasons, such as conflicts related to the numerous taxa with a reduced number of scored characters (missing data) and/or to the scarcity of overlapping elements among taxa (e.g., various specimens have few elements preserved), or still to a high ratio of conflicting information. To evaluate the causes of conflict in the postcranial dataset we ran an analysis using the TNT script IterPCR (Pol & Escapa, 2009). The results (Supplemental Information) indicate that the main cause of conflict in this dataset is missing data. Results show that for 25 unstable taxa (out of 35) the instability is caused by missing data. Still, for only 10 of these 35 the instability is related to both missing data and conflicting information among different characters (i.e., character states of distinct characters indicating alternative and controversial positons). Accordingly, in order to better explore the data, we pruned the most unstable taxa of the MPT’s of this analysis by using the command pcrprune in TNT (Goloboff & Szumik, 2015).

Figure 15 Reduced strict consensus tree of the analysis based only on postcranial characters after the exclusion of very unstable taxa.

Clades identified with a white circle represent informal clades. Taxa marked with * have a seemingly anomalous position within each informal clade recovered. Silhouettes of representative crocodylomorphs from Bronzati, Montefeltro & Langer (2012) and Bronzati, Montefeltro & Langer (2015).

In the reduced strict consensus (Fig. 15), Notosuchia is recovered with a similar taxonomic content as in the original analysis (i.e., including peirosaurids, uruguaysuchids and ziphosuchians). However, the relationship between peirosaurids and uruguaysuchids, as well as among some other notosuchians, differ from the original results (Leardi, Fiorelli & Gasparini, 2015). Yet, the importance of postcranial morphology to support the affinities of peirosaurids to notosuchians is strengthened, following previous evidences presented by Pol et al. (2012) and Pol et al. (2014). Also, the presence of a monophyletic Notosuchia illustrates the peculiarity of the notosuchian postcranial anatomy, which could be related to the emergence of a new terrestrial lifestyle, different from other terrestrial crocodyliforms, such as the “protosuchians.” Further, the results of the analyses using only the postcranial information show that some “protosuchians” are found together with the notosuchians, in a clade with only terrestrial forms (the only exception being Leidyosuchus and the affinity of this taxa to the terrestrial forms is derived from characters based on osteoderm anatomy). The Thalattosuchia clade is also recovered in this analysis, illustrating the peculiar postcranial anatomy of these taxa linked to a fully aquatic lifestyle. Another clade recovered includes semi-aquatic crocodyliforms (the only exception being Shamosuchus), including goniopholidids and eusuchians, but their relations largely deviate from the “control analysis.”

Overall, the results of these exploratory analyses indicate that crocodyliform relationships are strongly determined by skull characters. The postcranium has its importance in defining some relationships (i.e., those that appear in the control and original analyses but not in the analysis with cranial characters only), such as the affinity of peirosaurids and uruguaysuchids to Notosuchia, the monophyly of sebecosuchia (in the context of the original dataset used here). However, the general arrangement is still determined by characters related to the skull.

Finally, we do not consider that the results presented here reflect the inability of postcranial data to illustrate the evolutionary history of the group. Indeed, we consider that this is influenced by historical factors associated with the study of fossil crocodyliforms. Descriptions are usually based on skulls; postcranial elements are neglected, sometimes never described or mentioned in the descriptive works. However, the postcranium may play a bigger role in phylogenetic studies, as Crocodyliformes range from fully terrestrial animals to semi-aquatic and fully marine forms, and this diversity in lifestyle leads to different postcranial morphologies (e.g., Riff & Kellner, 2011; Molnar et al., 2015). Indeed, our exploratory analysis performed only with postcranial characters recovered three clades mainly representative of three different lifestyles (a “terrestrial” clade, a “semi-aquatic” clade, and a “marine” clade). However, the different homoplasy indexes show that this grouping is probably not a result of convergent events. The Rescaled Consistency Index (RCI—Farris, 1989) for the analysis with postcranial characters is 0.37, higher than those for the analyses with cranial characters (0.28), the control analysis (0.28), or the original analysis (0.22). A direct comparison of these values might be misleading, as different datasets exhibit particularities that could influence the results. For example, the higher RCI value for the postcranial dataset could result from the high percentage of missing data, as data of this nature cannot be homoplasious (71% in the postcranial dataset, against 37% in the cranial dataset, 47% in the control dataset, and 55% in the original dataset). On the other hand, this great number of missing data in the postcranial data set also suggests that there is still much to explore on the postcranial anatomy of Crocodyliformes, as the amount of missing data is not only related to the absence of preserved materials but also because studies describing postcranium are scarce. In this way, future work, describing more postcranial elements and proposing more characters based on this type of data will show if the phylogeny of Crocodyliformes is truly “skull-based” or merely “skull-biased.”

Conclusions

The study of the postcranial skeleton of Pissarrachampsa sera allowed the recognition of some exclusive features of this taxon in the context of Baurusuchidae, such as the short and sharp crest at the craniolateral margin of the distal tibial expansion, the raised and proximodistally elongated iliofibularis trochanter of the fibula, and the more proximally placed contact between the fibular distal hook and the tibia. Also, some features related to a terrestrial lifestyle were identified, as the reduced interosseous space between both radio-ulna and tibia-fibula, the tubercle in the lateral surface of the ischium, as well as a well-protruded medial facet and a well-excavated fossa flexoria in the tibia.

An important feature is the complete absence of osteoderms in Pissarrachampsa sera, the first suggested for a terrestrial crocodyliform. This complete loss of body armor was previously known only for metriorhynchids, which have extreme adaptations for a fully marine habit. In this scenario, osteoderms probably played a minor role in locomotion of terrestrial baurusuchids, with their complete absence in Pissarrachampsa sera representing the endpoint of this trend in the group. Further, the body size and mass estimations indicate that P. sera was a large predator in the terrestrial ecosystems of the Bauru Group, but it is unlikely that it fed on adult sauropods also present at this stratigraphic unit.

Finally, our exploratory phylogenetic analyses indicate that, at least for the matrix used in this study, crocodyliform relationships are determined primarily by skull characters. However, this is more likely a consequence of the high percentage of missing data in the postcranial data set and not of the inability of this data to reflect the evolutionary history of Crocodyliformes.

Supplemental Information

Supplemental Information 1 Body size and mass estimations and details of the phylogenetic analyses

Click here for additional data file.

Supplemental Information 2 Phylogenetic matrices

Matrices used for phylogenetic analyses in this study, including the exploratory analyses (nexus format).

Click here for additional data file.

We thank the staff of the Laboratório de Paleontologia de Ribeirão Preto (Universidade de São Paulo), specially Elisabete Gimenes Dassie, for the help with screening of specimens. We also thank Maíra Massarani that made part of the excavation crew. Access to comparative fossil specimens was possible thanks to Carl Mehling and Mark Norell (AMNH), Thiago Marinho (CPPLIP), Douglas Riff (UFU), William Simpson (FMNH), Rodrigo da Rocha Machado (DGM), Stella Alvarez and Alejandro Kramarz (MACN), Lorna Steel (NHMUK), Sheena Kaal (SAM), Patricia Holroyd and Kevin Padian (UCMP), and Liu Jun, Corwin Sullivan, and Zheng Fang (IVPP). Thorough reviews by Agustin Martinelli, James Clark, and Diego Pol greatly improved the final manuscript. This contribution used TNT v.1.1, a program made freely available thanks to a subsidy by the Willi Hennig Society.

Institutional Abbreviations

AMNH American Museum of Natural History, New York, USA.

CPPLIP Centro de Pesquisas Paleontológicas Llewellyn Ivor Price, Peirópolis, Uberaba, Brazil.

FMNH Field Museum of Natural History, Chicago, Illinois, USA.

DGM Museu de Ciências da Terra, Departamento Nacional de Produção Mineral (DNPM), Rio de Janeiro, Brazil.

IVPP Institute of Vertebrate Paleontology and Paleoanthropology, Chinese Academy of Sciences, Beijing, China.

LPRP/USP Laboratório de Paleontologia de Ribeirão Preto, Universidade de São Paulo; Ribeirão Preto, Brazil.

MACN Museo Argentino de Ciencias Naturales, Buenos Aires, Argentina.

MUCP Museo de Geología y Paleontología, Universidad Nacional del Comahue, Neuquén, Argentina.

MZSP Museu de Zoologia da Universidade de São Paulo, São Paulo, Brazil.

NHMUK Natural History Museum, London, UK.

SAM Iziko-South African Museum, Cape Town, South Africa.

UA University of Antananarivo, Antananarivo, Madagascar.

UCMP University of California Museum of Paleontology, Berkeley, California, USA.

UFRJ Museu de Paleontologia e Estratigrafia, Universidade Federal de Rio de Janeiro, Rio de Janeiro, Brazil.

UFU Universidade Federal de Uberlândia, Uberlândia, Brazil.

Additional Information and Declarations

Competing Interests

Author Contributions

Data Availability

The authors declare there are no competing interests.

Pedro L. Godoy and Mario Bronzati conceived and designed the experiments, performed the experiments, analyzed the data, contributed reagents/materials/analysis tools, wrote the paper, prepared figures and/or tables, reviewed drafts of the paper.

Estevan Eltink, Júlio C. de A. Marsola and Giovanne M. Cidade contributed reagents/materials/analysis tools, wrote the paper, prepared figures and/or tables.

Max C. Langer contributed reagents/materials/analysis tools, reviewed drafts of the paper.

Felipe C. Montefeltro conceived and designed the experiments, analyzed the data, contributed reagents/materials/analysis tools, wrote the paper, prepared figures and/or tables, reviewed drafts of the paper.

The following information was supplied regarding data availability:

The raw data is available as Supplemental Information.

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
