# Peer review of "Postcranial anatomy of Pissarrachampsa sera (Crocodyliformes, Baurusuchidae) from the Late Cretaceous of Brazil: insights on lifestyle and phylogenetic significance"

_PeerJ, doi:10.7717/peerj.2075_

## Round 0.1 · original submission · Minor Revisions

Dear authors,

I have accepted the decision of 'minor revision' from the reviewers.

I have some additional comments that the authors should address prior to resubmission (in addition to those made by the reviewers):
1. Authority and date should be provided for each species-level taxon at first mention. Please ensure that the nominal authority is also included in the reference list.
2. Please replace 'trees' with 'cladograms' where appropriate.
3. In the discussion section on ‘Body size and mass estimates for Pissarrachampsa sera’, you discuss using the Alligator femoral regression questions to estimate potential body size. Another reference that demonstrates different scaling ratios exist for crocodylomorph taxa that deviate from extant morphotypes include: Young et al. (2011), ZJLS 163: 1199-1216.

·

Basic reporting

The authors have done an excellent work describing the postcranial anatomy of the baurusuchid Pissarrachampsa sera with broad discussions on different aspects.
The detailed description and comparisons, the discussed issues (corporal size, lack of osteoderms, etc.) and the different analyses testing the impact of postcranial features among Crocodyliformes phylogeny make this MS an important contribution to vertebrate paleontology.

There are several comments I have made along the text pdf file that should be followed by the authors.

Some particular issues are:

--Bauru Group is located along southeastern Brazil and not southern or southwestern Brazil.
--There are several problems about the correlation of the Bauru Group units, as stated by the authors, and also, the inferred age is quite problematic.
--The discussion on baurusuchids vs theropods “competition” was also commented in other published contributions that are not mentioned in the text.
--The references for theropod record at Bauru Group are quite limited (see comments).
--About the lack of osteoderms: It is very interesting the condition in P. sera, especially for two reasons: its basalmost position within Baurusuchidae and that it is a smaller taxon in comparison with, for example, Stratiotosuchus and Baurusuchus. I think I should be commented in the text.
-- The scale bar of Figure 12 seems to be extremely large. I think it is concomitant with the estimated >3 meter long Croc based on Farlow et al., (2005). But the scale seems to be larger for real size of the bones. Is that Ok?
--See other comments along the Ms.

Experimental design

The anatomical descriptions and phylogenetic studies are based on reliable analyses, comparisons and techniques.

Validity of the findings

No comments

Additional comments

See Basic Reporting and the attached pdf file with comments.

·

Basic reporting

This is a detailed description of the postcranial skeleton of Pissarrachampsa sera, a terrestrial carnivorous crocodyliform from the Late Cretaceous of Brazil. Postcrania are not as well known as crania for fossil crocodyliforms, so this is an important addition to our knowledge of this group. I have made many notes on the file of the text, but the description is basically sound. There needs to be more information on the relationships of the specimens to one another in situ, and whether any of the referred specimens could belong to the same taxon. The authors should consult Irmis (2007, Journal of Vertebrate Paleontology 27: 350–361) on variability in vertebral sutures, and Bailleul et al. (2015, PLoS ONE 11(2): e0147687) on difficulties in identifying the adult stage of crocodyliforms. I am confused by the contention that the femoral head of extant crocodylians is more inturned than that of Pissarrachampsa, since the former do not have an erect posture while the latter is inferred to have a vertical femur here. Osteoderms are also reported as absent in the sphenosuchian Junggarsuchus, and this was somewhat confirmed by the recent discovery of a similar taxon from the Daohugou beds without osteoderms (Sullivan presented a talk at the last SVP meeting on this). The references to figures need to give not only the number but the letter to the part of the figure.

Experimental design

The design of the experiments is fine.

Validity of the findings

The phylogenetic analyses are fine, although the wording needs attention. I am confused by the statement that Thalattosuchia are neosuchians in the postcranial analysis, the cladogram does not show that. The authors are probably correct that the higher RI is due to missing data in the postcranial data set, so they should give the percentage of missing data for each of the sets. The discussion of terrestriality needs to explain better why some of the features are related to cursoriality, for example there is no explanation for why a tubercle on the ischium for the M. pubioischiotibialis is related to upright posture. The discussion of the lack of osteoderms should also comment on whether there are features of the vertebrae that indicate a lack of the suspensory system involving the osteoderms that has been described by Frey.

Additional comments

This is basically a fine paper that will be an addition to our knowledge of fossil crocodyliforms, but there are a few minor issues that need to be dealt with.

·

Basic reporting

The article is well written and I only have noted minor edits in this regard in the Word Document (attached to this review).

The article is sound and includes an adequate introduction.

The figures are very well executed and clearly shows the described morphology.

As far as I can tell, this submissions meets the PeerJ standards

Experimental design

The research fits well within the scope of the journal and provides new data and analysis on the anatomy and evolution of the postcranium of a Cretaceous crocdyliform.
As noted by the authors there is a significant gap in this topic and this manuscript helps filling this gap.
The description is very well done, clear and with adequate comparisons. It will be a useful contribution for all researchers interested in anatomy of Crocodyliformes.

Validity of the findings

I only have minor comments regarding the description, but I have made several comments on the discussion.

The Discussion deals with very important topics and are tightly relevant to the described anatomy. All the sections of the discussion could be improved and I have suggested possible lines of development for them (especially the exploratory analyses conducted to evaluate the impact of postcranial characters in the evoution of Crocodyliformes).

Nothing of what the authors have stated is wrong or wrongly conducted, but my comments point to possible lines through which the authors can expand and further explore these topics.

---

## Round 0.2 · Minor Revisions

Dear authors,

Thank you for your prompt return of the manuscript. Reviewer two has some minor edits that should be addressed before acceptance.

·

Basic reporting

The authors have addressed all comments and suggestions made in the original MS. With these modifications and those suggested by other reviewers, the MS was considerably improved.
I agree that this version should be considered for publication in PeerJ.

Experimental design

No comments at this version (R1)

Validity of the findings

No comments at this version (R1)

Additional comments

All suggestions were followed. I recommend to accept the Ms.

·

Basic reporting

Fine

Experimental design

Fine

Validity of the findings

Fine

Additional comments

The revisions mostly address my comments in the earlier review (my thalattosuchians-in-Neosuchia comment was due to reading the wrong figure legend). The one that is not fully addressed is how the postcranial specimens were preserved relative to the cranium; the authors just need to explain how close they were (a few centimeters or several meters?). ("this association is possible as the postcranial elements were spacially identified during the first expedition, when the holotypic skull was collected.") I've made some small edits on the attached file.

---

## Round 0.3 · accepted · Accept

Dear authors,

I have looked over your revised manuscript, and I am satisfied with your response to the reviewers comments.